



# The demise of the world's largest piedmont glacier: a probabilistic forecast

Douglas J. Brinkerhoff[1], Brandon S. Tober[2,3], Michael Daniel[3], Victor Devaux-Chupin[4], Michael S. Christoffersen[3,4], John W. Holt[3], Christopher F. Larsen[4], Mark Fahnestock[4], Michael G. Loso[5], Kristin M. F. Timm[6], Russell C. Mitchell[1], and Martin Truffer[4]

[1]Department of Computer Science, University of Montana, Missoula, MT, USA
[2]Department of Civil and Environmental Engineering, Carnegie Mellon University, Pittsburgh, PA, USA
[3]Department of Geosciences, University of Arizona, Tucson, AZ, USA
[4]Geophysical Institute, University of Alaska Fairbanks, Fairbanks, AK, USA
[5]Wrangell–St. Elias National Park and Preserve, National Park Service, Copper Center, AK, USA
[6]International Arctic Research Center, University of Alaska Fairbanks, Fairbanks, AK, USA

**Correspondence:** Douglas J. Brinkerhoff (doug.brinkerhoff@mso.umt.edu)

**Abstract.** Sít' Tlein in Alaska's St. Elias Range is the world's largest piedmont glacier and has thinned considerably over 30 years of altimetry, yet it's low-elevation piedmont lobe has remained intact in contrast to the glaciers that once filled neighboring Icy and Disenchantment bays. In an effort to forecast changes to Sít' Tlein over decadal to centennial time scales, we take a data-constrained dynamical modelling approach, in which we constrain the parameters of a higher order model of ice flow – the bed elevation, basal traction, and surface mass balance – with a diverse but spatio-temporally sparse set of observations
including satellite-derived time-varying velocity fields, radar-derived bed and surface elevation measurements, and *in situ* and remotely sensed observations of accumulation and ablation. Nonetheless, such data do not uniquely constrain model behavior, so we adopt an approximate Bayesian approach based on the Laplace approximation and facilitated by low-rank parametric representations to quantify uncertainty in the bed, traction, and mass balance fields alongside the induced uncertainty in model-
based predictions of glacier change. We find that Sít' Tlein is considerably out of balance with contemporary (and presumably future) climate, and we expect its piedmont lobe to largely disappear over the coming 150 years. We forecast a total mass loss at Sít' Tlein of between 500 and 1000 km$^3$ of ice, a range that represents not only uncertainty in model inputs, but also in future warming scenarios. The resulting retreat and subsequent replacement of glacier ice with a marine embayment or lake will yield a significant modification to the regional landscape and ecosystem.

## 15  1  Introduction

Sít' Tlein (briefly known as Malaspina Glacier, Fig. 1a), situated in coastal Alaska in the St. Elias Mountains, is the world's largest piedmont glacier, and when taken together with its neighbor the Bering-Bagley icefield is Earth's largest temperate ice mass. Its geometry is complex and is comprised of a large piedmont lobe that is fed by three principal tributaries. The largest of these tributaries (sometimes independently referred to as the Seward Glacier) has in its accumulation area Mts. Logan and St.
Elias, the second and fourth highest points in North America, while its smaller tributaries, the Agassiz and Marvine Glaciers,





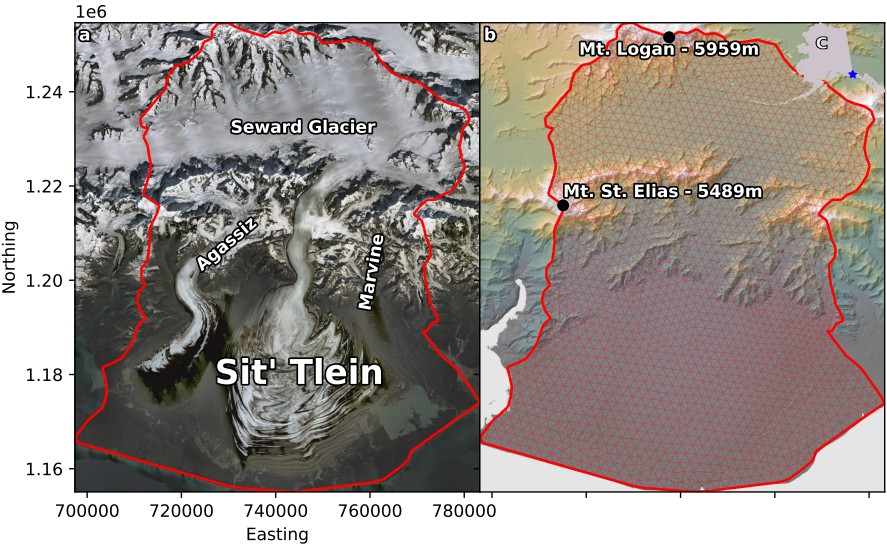

**Figure 1.** Sít' Tlein lobe and its tributary basins (a). A digital elevation model alongside our 1.5km resolution finite element mesh (b). The location of Sít' Tlein within the US state of Alaska (c). The red outline indicates the model domain.

transport ice from the maritime windward slopes of Mts. St. Elias and Cook to within a few kilometers of the Gulf of Alaska. The collected area of Sít' Tlein and its tributaries is roughly 4500 km², and the piedmont lobe has an estimated volume of nearly 700 km³ (Tober et al., 2023). The lobe's volume is constrained by radar observations of the ice geometry (Tober et al., 2023), but the volumes of the tributaries are not well known.

Sít' Tlein is also thinning and the glacier extent has diminished since it was first reliably mapped in the late 19th and early 20th century (Russell, 1893; Tarr and Martin, 1914; Sharp, 1958), with the active ice front, which previously extended to the ocean in some locations, now typically more than a kilometer removed. The combined system exhibits complex dynamics, with both the tributaries and piedmont lobe undergoing periodic surges that transport ice to regions that are otherwise stagnant, and as such are critical for maintaining the piedmont lobe's geometry. These surges are spatially variable, with alternating

directionality leading to dramatic looped moraines (Muskett et al., 2008). Notably, the lobe was at one point contiguous with piedmont glaciers that filled adjacent Icy and Disenchantment Bays (Barclay et al., 2001). This is reflected in the Tlingit name Sít' Tlein, which translates to 'Big Glacier' and is also used to describe the extant Hubbard Glacier (Thornton, 2012), with its tidewater terminus located at the head of Disenchantment Bay. The conspicuous difference in retreat history between Sít' Tlein and its neighbors leads to the principal objective of this paper, which is to predict the evolution of Sít' Tlein - and in particular

the potential future disintegration of its piedmont lobe.

Our principal approach is through data-calibrated modelling. We use the ice dynamics model SpecEIS (Brinkerhoff, 2022, described in Sec. 2) – which we denote $\mathcal{M}$ – to explicitly evolve the ice thickness $H(\mathbf{x},t)$ and velocity field $\mathbf{u}(\mathbf{x},z,t)$ of Sít' Tlein from the year 1915 until 2344 based on past and assumed future climate forcings. While physics-based models are a




useful tool for answering questions about glacier evolution under different assumptions of climate forcing, ice flow models

in general and SpecEIS in particular are dependent on several critical inputs that govern the model's behavior. These inputs – so-called *parameters* – include the elevation of the glacier bed $B(\mathbf{x})$, the spatio-temporally distributed frictional properties governing sliding at the glacier base $\beta(\mathbf{x},t)$ (changes in which presumably control the observed surging at Sít' Tlein), the spatio-temporally distributed specific surface mass balance rate $\dot{a}(\mathbf{x},t)$, as well as an initial ice thickness $H_0(\mathbf{x})$ from which to begin the time-evolution of the glacier geometry. Taken together, these form the vector

$$\mathbf{m} = [H_0(\mathbf{x}), B(\mathbf{x}), \beta(\mathbf{x},t), \dot{a}(\mathbf{x},t)]. \tag{1}$$

Each of these parameters exerts a significant control on glacier evolution. At Sít' Tlein (and elsewhere), we do not have a full characterization of $\mathbf{m}$ over space and time, which is a significant impediment to reliable projections of glacier change; however, *some* of its elements are partially observed at discrete locations and times. In this case we have spatiotemporally sparse radar-derived observations of the bed elevation $\hat{B}$ and surface mass balance $\hat{a}$. Such parameters also indirectly influence

remotely sensed observations of surface velocity $\hat{u}$ and aerial laser altimetry of surface elevation $\hat{S}$. These observations together form the vector of observations

$$\mathbf{d} = [\hat{S}, \hat{\mathbf{u}}_s, \hat{B}, \hat{a}]. \tag{2}$$

We seek to use these observations to constrain – to the extent possible – the model's parameters such that when used in conjunction with the ice flow model to predict glacier evolution over the period for which observations exist, predictions are

consistent with observations. In practice, because these parameters are continuous functions in space and time, we must make assumptions about how to represent them so as to be representable on a computer and to exhibit feasible physical properties such as smoothness. The construction of these representations is the topic of Sec. 3.

Because our observations are both imperfect and sparse, it is not possible (nor desirable) to identify a single ideal model configuration, and as such we adopt a probabilistic approach to prediction. Given a quantity of interest – which we call $\Delta(t)$,

and which could represent change in elevation at a point, total ice volume, changes in meltwater flux, or any other model-derived quantity – we seek to characterize the probability distribution

$$P(\Delta(t)|\mathbf{d}, \mathcal{F}, \mathcal{M}) = \int P(\Delta(t)|\mathbf{m}, f, \mathcal{M}) \times$$
$$P(\mathbf{m}|\mathbf{d}, \mathcal{M}) \, P(f|\mathcal{F}) \, \mathrm{d}\mathbf{m} \, \mathrm{d}f, \tag{3}$$

where $P(\cdot|\cdot)$ denotes a probability density function over the first argument, with the second argument representing given

conditions. This distribution can be interpreted in the sense of an ensemble of simulations of future change – ensemble members are drawn from the distribution of exogenous climate forcings $P(f|\mathcal{F})$ that are plausible under a chosen future climate scenario $\mathcal{F}$ alongside endogenous (but data-constrained) parameters drawn from $P(\mathbf{m}|\mathbf{d}, \mathcal{M})$.

As is typical for probabilistic prediction, we characterize Eq. 3 – the predictive distribution – in two steps. In the first step, described in Sec. 4 we infer the distribution of model parameters at Sít' Tlein given observations, i.e. we solve an 'inverse





problem'. This corresponds to finding the distribution over $\mathbf{m}$ such that resulting hindcasts over the historic period from 1915 to 2023 agree with available observations. This distribution – which we call the posterior – can be expressed as

$$P(\mathbf{m}|\mathbf{d},\mathcal{M}) \propto \underbrace{P(\mathbf{d}|\mathbf{m},\mathcal{M})}_{\text{likelihood}} \underbrace{P(\mathbf{m}|\mathcal{M})}_{\text{prior}}, \tag{4}$$

where we have used Bayes' theorem to write the posterior distribution as proportional to the product of a likelihood term - which measures the degree to which predictions made given a particular set of model parameters agree with the observations in hand - and a prior - which measures how likely said parameters were before observational constraint. Because our parameters are high-dimensional and our flow model nonlinear, characterizing this distribution exactly is not possible. To partially circumvent this we describe a numerical approximation method based on a local quadratic expansion and randomized low-rank matrix decomposition.

In the second step, described in Sec. 5, we approximate Eq. 3 under a handful of assumptions about future climate and calving dynamics by drawing a finite collection of random samples over future forcings and model parameters from the posterior distribution Eq. 4 and using SpecEIS to predict a range of plausible glacier changes from present to 2344, with a particular emphasis on assessing the stability of the piedmont lobe.

## 2 Ice dynamics model

The posterior distribution over parameters is conditioned on a choice of model $\mathcal{M}$. This conditioning specifies which model parameters need to be inferred, and also specifies – through the physical processes that the model represents – the way that a particular choice of parameter value is translated into something that can be compared against observations via the likelihood model.

Here we model glacier dynamics using the ice flow model SpecEIS (Spectral Element Ice Simulator Brinkerhoff, 2023), which solves the coupled equations of mass conservation and stress balance defined over a domain $\Omega$ with boundary $\Gamma$. Mass conservation is expressed through the continuity equation

$$\frac{\partial H}{\partial t} + \nabla_{\mathbf{x}} \cdot \bar{\mathbf{u}}H = \dot{a}, \quad H > 0 \qquad \text{on } \Omega \tag{5}$$

$$(\bar{\mathbf{u}}H) \cdot \mathbf{n}_{\mathbf{x}} = q_{in}, \qquad \text{on } \Gamma_{in} \tag{6}$$

where $\mathbf{u}(\mathbf{x},z,t)$ is the ice velocity, $\bar{\mathbf{u}}(\mathbf{x},t)$ is its vertical average, $H(\mathbf{x},t)$ is the ice thickness, and $q_{in}$ is a boundary flux (which we henceforth take to be zero). The stress balance (here the Blatter-Pattyn approximation (BPA) to the Stokes equations Pattyn, 2003) is

$$\nabla_{\mathbf{x},z} \cdot 2\eta\dot{\epsilon}_1 = \rho_i g \nabla_{\mathbf{x}} S, \qquad \text{on } \Omega, \tag{7}$$





where $\rho_i$ and $g$ are ice density and gravitational acceleration, $S(\mathbf{x},t) = z_B(\mathbf{x},t) + H(\mathbf{x},t)$ the surface elevation, and $\nabla_{\mathbf{x},z} \equiv \left[\frac{\partial}{\partial x_1}, \frac{\partial}{\partial x_2}, \frac{\partial}{\partial z}\right]^T$. The elevation of the ice base is

$$z_B(\mathbf{x},t) = \max\left(z_{sl} - \frac{\rho_i}{\rho_w}H(\mathbf{x},t), B(\mathbf{x})\right), \tag{8}$$

with $z_{sl}$ the sea level. As such, (7) applies to both grounded and floating ice. $\dot{\epsilon}_1$ is the strain rate tensor subject to the simplifications of the BPA:

$$\dot{\epsilon}_1 = \begin{bmatrix} \left(2\frac{\partial u_1}{\partial x_1} + \frac{\partial u_2}{\partial x_2}\right) & \frac{1}{2}\left(\frac{\partial u_1}{\partial x_2} + \frac{\partial u_2}{\partial x_1}\right) & \frac{1}{2}\frac{\partial u_1}{\partial z} \\ \frac{1}{2}\left(\frac{\partial u_1}{\partial x_2} + \frac{\partial u_2}{\partial x_1}\right) & \left(\frac{\partial u_1}{\partial x_1} + 2\frac{\partial u_2}{\partial x_2}\right) & \frac{1}{2}\frac{\partial u_2}{\partial z} \end{bmatrix}. \tag{9}$$

The viscosity – which depends inversely on the effective rate of strain and thus describes a shear-thinning fluid – is given by Glen's flow law

$$\eta = \frac{1}{2}A^{-\frac{1}{n}}(\dot{\epsilon}_{II}^2)^{\frac{1-n}{2n}}, \tag{10}$$

with $A$ the ice hardness, $n$ the flow exponent and $\dot{\epsilon}_{II}$ the second invariant of the strain rate tensor. At subaerial lateral boundaries, we assume a stress free condition, while at subaqueous boundaries we assume a normal stress given by water pressure. At the interface between ice and substrate, we assume a Budd-type sliding law (Budd et al., 1979)

$$2\eta\dot{\epsilon} \cdot \mathbf{n} = -\exp(\beta)N\mathbf{u}, \tag{11}$$

where $N$ is the effective pressure, with water pressure assumed to be the maximum of 80% of the ice overburden pressure or the sea-level induced water pressure.

## 2.1 Discretization

SpecEIS discretizes in space using a mixed finite element method. Finite-element methods represent spatial functions as a linear combination of fixed basis functions that are defined on a mesh, which we take here to be a triangular tesselation of the model domain (Fig. 1b); each triangle in the mesh is called a cell, and in SpecEIS we represent the thickness field as a weighted sum of cell-wise constants (e.g. as a function in the zero-order Discontinuous Galerkin space, hereafter abbreviated DG0, Boffi et al., 2013).

$$H(\mathbf{x},t) \approx \sum_{k \in |\mathcal{T}|} \phi_k(\mathbf{x})\mathsf{H}_k(t), \tag{12}$$

where $\mathcal{T}$ is the set of mesh cells,

$$\phi_k(\mathbf{x}) = \begin{cases} 1 \text{ if } \mathbf{x} \in \mathcal{T}_k \\ 0 \text{ else}, \end{cases} \tag{13}$$



and $\mathsf{H}_k(t)$ is the time-varying thickness coefficient, which here can reasonably be interpreted as the average thickness across mesh cell $k$. The velocity is similarly represented as a weighted sum of basis functions

$$\mathbf{u}(\mathbf{x},z,t) \approx \sum_{k \in |\mathcal{E}|} \sum_{l=1}^{3} \boldsymbol{\psi}_{kl} \left[ \bar{\mathsf{U}}_{kl} + \frac{\mathsf{U}'_{kl}}{n+1}((n+2)\varsigma^{n+1} - 1) \right], \tag{14}$$

where $\mathcal{E}$ is the set of mesh edges, $\boldsymbol{\psi}_{kl}(\mathbf{x})$ is the $l$-th Mardal-Tai-Winther basis function (Mardal et al., 2002, of which there are three per edge) associated with the $k-$th edge, and $\bar{\mathsf{U}}_{kl}$ and $\mathsf{U}'_{kl}$ are coefficients associated with the depth-averaged and depth-varying components of the ice velocity (the Monolayer Higher-Order approximation, Dias dos Santos et al., 2022). In contrast to the thickness discretization, these velocity coefficients represent the magnitude of the velocity field normal and tangential to cell edges. This combination of finite elements generalizes the so-called Arakawa staggered C-grid (Arakawa and Lamb, 1977) that is frequently used for shallow-ice models to correctly account for the longitudinal stresses of the Blatter-Pattyn approximation, and is known to maintain thickness positivity and uphold mass conservation while being free from spurious numerical wiggles. SpecEIS discretizes in time using the backward Euler method *applied simultaneously to both equations*, and as such is fully implicit. The resulting non-linear system of equations is solved using a damped Picard iteration. A detailed description of this model, as well as a full suite of experiments quantifying and verifying model performance is found in Brinkerhoff (2023).

Internal to the model, we maintain a finite element representation of the parameters $\mathbf{m}$. $H_0(\mathbf{x})$ is represented identically to $H(\mathbf{x},t)$, and we refer to its coefficients as $\mathsf{H}_0$. The surface mass balance and bed elevation are also represented using the DG0 space, with coefficients $\dot{\mathsf{a}}$ and $\mathsf{B}$ respectively. In contrast, the basal traction uses a first-order continuous Galerkin (piecewise linear, CG1) basis with coefficients $\beta$ associated with a nodal basis.

We discretize the contributing area of Sít' Tlein at a relatively coarse 1.5km horizontal resolution (although we note that experimentation with higher resolution meshes does not yield significantly different results). The resulting mesh has 3898 cells, 1997 nodes, and 5824 edges. While it is possible in this framework to adapt mesh element sizes with respect to a desired criterion (commonly velocity magnitudes, strain rate magnitudes, or grounding line proximity), we expect both the velocity and geometry to change significantly over the course of our simulations so we opt for a nearly uniform element size distribution under the assumption that this mesh is of sufficient resolution to capture the glacier's essential features.

## 2.2 Integration with Pytorch

Because we seek to perform statistical inference and optimization using this model, we require the derivatives of (scalar functions of) the model outputs with respect to its inputs, i.e. the gradient of the average model error with respect to a parameter. At its simplest, an ice flow model can be written in a fully discrete form for a single time step (using the Markov property of the equations) as

$$[\mathsf{H}, \bar{\mathsf{U}}, \mathsf{U}'] = \mathsf{SpecEIS}(\mathsf{H}_0, \mathsf{B}, \beta, \dot{\mathsf{a}}; \Delta t), \tag{15}$$

for some time step size $\Delta t$. In this representation, inputs and outputs are both just arrays of numbers, and the resulting model becomes amenable to inclusion in a general purpose reverse-mode automatic differentiation (AD) framework such as Pytorch





(Paszke et al., 2019). In the parlance of Pytorch, Eq. 15 constitutes a *forward* function. To implement a *backward* function we require a routine that efficiently computes the product of a vector with the Jacobian of the SpecEIS function. We use the adjoint
method (e.g. MacAyeal, 1993; Heimbach and Bugnion, 2009, in the glaciological literature) to evaluate such vector-Jacobian products (See Appendix A). This discrete and modular view is powerful because the thickness of the ice at the beginning of the time step is a function argument that will have a gradient associated with it when reverse-mode AD is applied. Reverse-mode AD generally, and Pytorch specifically, support arbitrary function composition, we can arbitrarily compose this discrete function with other functions. As we will see, these can be either complex routines for characterizing misfit which will facilitate
complex statistical treatment, or the function itself. The latter yields a fully time-dependent adjoint that can help determine the sensitivity of the model at the end of a simulation to parameter values at all times – for example the sensitivity of the average surface elevation error through time to a surface mass balance applied long in the past.

## 3   Representation of model parameters

The parameters in $\mathbf{m}$ to be nominally inferred – the bed elevation $B(\mathbf{x})$, the basal traction $\beta(\mathbf{x}, t)$, and surface mass balance
$\dot{a}(\mathbf{x}, t)$ – are each complex and continuous functions of space and (perhaps) time. As such, we introduce an approximating probabilistic model (i.e. a prior distribution) for each of these parameters which we then make amenable to computer representation via decomposition into a finite set of basis functions. This representation is separate from the finite element bases described above, a distinction that is necessary because the characteristics of a finite element mesh do not necessarily impose desired physical properties. However, we also introduce the necessary mechanisms for mapping samples from this basis to the
finite element mesh. In introducing these parameter models, we necessarily make some assumptions about smoothness and characteristic scales of variability, while also making our representation independent of the numerical treatment of the model. This latter point is important because it allows for a natural hierarchy of mesh refinement in which some computational tasks can be performed with a coarsely defined ice flow model, whereas others can be performed with more detail. We emphasize that any explicit functional representation of model parameters is subject to potential mis-specification (as is the flow model
itself). However, doing so is also unavoidable so we endeavor to be as transparent as possible about such assumptions.

Each parameter in $\mathbf{m}$ is characterized by a different type of spatio-temporal variability; we describe our approach to modelling each below, which is equivalent to establishing a prior distribution. In addition to the specification of functional forms, it is also convenient to condition these priors (via Bayes rule) when direct observations (i.e. observations that need not be used in conjunction with the ice flow model) are available. These data-constrained distributions, by virtue of our choice of prior
family and data likelihood, are analytically tractable and are treated as an updated prior from the perspective of more complex inference involving SpecEIS.

### 3.1   Bed elevation

We parameterize the probability distribution over the bed elevation as a Gaussian process (GP Williams and Rasmussen, 2006, from whom we also adopt notation) in space, which assumes that the function value at any two coordinates $\mathbf{x}$ and $\mathbf{x}'$





are jointly normal, with a covariance given by a kernel function $k(\mathbf{x}, \mathbf{x}')$ and mean function $m(\mathbf{x})$. Throughout this and the following sections, we use this notation frequently for different parameters and do not differentiate between them for concision of presentation – we hope that which parameter we refer to is clear from context. Evaluated at a finite set of spatial points $X \in \mathbb{R}^{n \times 2}$ (for example, the finite set of observation locations or the quadrature points of a finite element mesh), we can describe the distribution over this function with a normal distribution

$$P(B(X)) = \mathcal{N}(\boldsymbol{\mu}, K), \tag{16}$$

where $K \in \mathbb{R}^{n \times n}$ is the covariance matrix $K_{ij} = k(X_i, X_j)$, and $\boldsymbol{\mu} = m(X) \in \mathbb{R}^n$ the mean vector. While unrealistically simplistic for representing the extreme and glacierized topography of the St. Elias range, we nonetheless use the Matern family of covariance functions. The behavior of the distribution is governed by a length scale, amplitude scale, and differentiability index. At Sít' Tlein, based on maximum marginal likelihood estimation Tober et al. (2023) we take the characteristic length

scale $l$ as 3km, the differentiability $\nu = \frac{3}{2}$, and the amplitude to be 1000 m. We model the distribution over the mean function as a polynomial

$$m(\mathbf{x}) = \mathbf{h}(\mathbf{x})^T \boldsymbol{\alpha}, \tag{17}$$

where $\mathbf{h}$ are a set of orthogonalized degree-two polynomial basis functions and $\boldsymbol{\alpha}$ is a coefficient vector. Taken together this model assumes that the topography is well approximated by a quadratic polynomial with local variability given by a GP.

### 3.1.1 Low rank decomposition

$K$ is often low-rank, which is to say that some of its columns contain redundant information (for example when points in $X$ are close together relative to the characteristic length scale of the GP). This property motivates a reparameterization of $P(B)$ as

$$B(X) = L\mathbf{w} + H\alpha$$
$$\mathbf{w} \sim \mathcal{N}(0, I),$$
$$\alpha \in \mathbb{R}^6, \tag{18}$$

where $H = \mathbf{h}(X) \in \mathbb{R}^{n \times 6}$ is a Vandermonde matrix and $L \in \mathbb{R}^{n \times r}$ is an approximate root of the spatial covariance matrix such that $LL^T \approx K$. Note that $L$ need not be square – if $K$ is indeed low rank, then $r$ may be much less than $n$. Under this reparameterization the degrees of freedom that characterize function behavior (and that need to be inferred in the inverse

context) are the coefficients $\mathbf{w} \in \mathbb{R}^r$, which are *a priori* unit-normal. Such a decomposition decouples the number of degrees of freedom necessary to represent the function (i.e. the length of $\mathbf{w}$) from the number of locations at which this function is to be evaluated, a desirable property. $B(X)$ under this choice of representation can be described as a combination of *finite* basis functions, and we refer to the matrix $L$ as the basis for $B(X)$.



### 3.1.2  Structured kernel interpolation

We now turn to the construction of $L(X)$. Many matrix decompositions produce an approximate matrix root; however, some choices are either intractable to compute, not low rank, or have undesirable numerical properties. The classic choice is a (truncated) eigendecomposition – however, this has two issues. First, it requires the explicit construction, storage, and manipulation of the matrix $K$. While randomized methods can circumvent some of the problematic scaling associated with computing the decomposition, circumventing the storage requirement is challenging. Second, despite this decomposition optimally capturing

the low rank structure of the target covariance matrix, it does not retain the characteristics of the underlying matrix; even though $K$ may be nearly sparse in the sense that the covariance between distant points is numerically zero, the associated basis has columns that are non-sparse – all things equal, we prefer a decomposition that retains the approximate sparsity of the original matrix.

Here we describe an approach that addresses both of these issues. In order to circumvent the requirement that we form

the matrix $K$ explicitly, we employ structured kernel interpolation (SKI Wilson and Nickisch, 2015) which posits that the covariance matrix can be approximated as

$$K \approx W(K^x \otimes K^y)W^T, \tag{19}$$

where $K^x$ and $K^y$ are 1D covariance matrices defined over regular grids in each map-plane dimension independently and $W$ an interpolation matrix. The Kronecker product of the two 1D covariance matrices is then a 2D covariance matrix evaluated on

regular grid. In this work, we take each of these 1D grids to extend a few length-scales past the boundaries of our mesh in each dimension with a spacing of $l/10$. The Kronecker product is not any easier to store explicitly; however, matrix-vector products can be efficiently computed by only forming each 1D covariance matrix independently.

This grid-evaluated covariance is not useful on its own – it provides correlations between function values at the wrong locations. To map this structured covariance to arbitrary locations $X$ we employ an interpolation matrix $W$ (in this work we

use inverse distance weighting). $W$ is highly sparse, with inverse distance weighting leading to only four non-zero entries per row (only the four grid points bounding the desired evaluation location contribute to the interpolation). As such the product of Eq. 19 with an arbitrary vector can be evaluated inexpensively. The error induced by this interpolation is typically small (Wilson and Nickisch, 2015). Again, while computing the left side of Eq. 19 is usually intractable, the computation and storage of each factor on the right side is straightforward, and their properties allow for the efficient computation of matrix-vector

products.

We can also efficiently product a matrix root. Again invoking the algebraic properties of the Kronecker product, we write

$$K \approx LL^T$$
$$L = W(L^x \otimes L^y), \tag{20}$$

where $L^x L^{xT} \approx K^x$ and similarly for $L^y$. Concretely, rather than decompose the matrix directly, we perform a component-

wise decomposition of separable Kronecker factored covariance matrices (which are very small), and then sparsely interpolate these to evaluation points.



As a final step, we must select a low-rank decomposition for the coordinate-wise covariance matrices. We use the Nyström approximation applied to a matrix root to compute the low-rank factor

$$L^x = K^x_{[:,\mathbf{p}]} V^x \Lambda^{x-\frac{1}{2}} V^{xT}), \tag{21}$$

where $\mathbf{p}$ are the indices of pivot columns (here selected rigorously through pivoted QR decomposition), and where $U^x$ and $\Lambda^x$ are the eigenvectors and values of $K^x_{[\mathbf{p},\mathbf{p}]}$ respectively. The resulting factor has as columns basis vectors that resemble entries in the covariance function, with the same approximate sparsity pattern, but which eliminates redundant basis elements and appropriately scales columns. The resulting procedure yields a representation with 4,192 degrees of freedom.

### 3.1.3 Conditioning on bed observations

Next we turn to constraining bed elevations from direct observations. NASA's Operation IceBridge collected over 4000 km of 2.5/5MHz radar soundings between 2013 and 2023, from which the glacier base can be interpreted with a nominal error of approximately 25m based on integrated analysis at sounding crossover points. A detailed analysis of this product can be found in (Tober et al., 2023). We combine these radar soundings – which are relevant for subglacial locations – with the Copernicus GLO-30 Digital Elevation Model (https://doi.org/10.5270/ESA-c5d3d65) masked to ice-free regions to produce a combined data set that provides observations of the bedrock elevation at varying degrees of spatial density across our study area. We assume that these observations are independent and normally distributed about the true bedrock elevation with observational standard deviation $\sigma_{obs} = 25$ m, or

$$P(\hat{B}|B(X_{\text{obs}})) = \mathcal{N}(B(X_{\text{obs}}), \sigma_{obs}^2 I), \tag{22}$$

where $X_{obs} \in \mathbb{R}^{n_{obs} \times 2}$ are the observation coordinates. We specify the coefficients of the mean function $\bar{\boldsymbol{\alpha}}$ by the standard least squares solution

$$\bar{\boldsymbol{\alpha}} = (H_{obs}^T H_{obs})^{-1} H_{obs}^T \hat{B}, \tag{23}$$

where $H_{obs} = \mathbf{h}(X_{obs})$. Because both this likelihood and also the prior distribution are normally distributed, and because the map from GP coefficients $\mathbf{w}$ to $B$ is linear, the posterior distribution has an analytical solution given by

$$P(\mathbf{w}|\hat{B}, \boldsymbol{\alpha}) = \mathcal{N}(\bar{\mathbf{w}}, \Sigma)$$
$$\Sigma = \left(I + L_{obs}^T \sigma_{obs}^{-2} L_{obs}\right)^{-1} \tag{24}$$
$$\bar{\mathbf{w}} = \Sigma L_{obs}^T \sigma_{obs}^{-2} \left(\hat{B} - H_{obs}\bar{\boldsymbol{\alpha}}\right), \tag{25}$$

and where $L_{obs}$ is given by the basis evaluated at observations points.

With this posterior distribution over the basis coefficients $\mathbf{w}$, we decompose the posterior covariance matrix, leading to the linear model

$$B(X) = L(C\mathbf{z} + \bar{\mathbf{w}}) + H\bar{\boldsymbol{\alpha}} \tag{26}$$
$$\mathbf{z} \sim \mathcal{N}(0, I) \tag{27}$$



with $C$ a root of $\Sigma$.

### 3.1.4 Mapping to the model grid

In order to use the bedrock elevation predictions of the Gaussian Process described above with the flow model we need to map

the modal basis coefficients $z_B$ to the finite element basis coefficients B, which are associated with the piecewise constant DG0 function space basis. One obvious way would be to evaluate Eq. 27 at mesh centroids. However, we observe that the DG0 basis has less regularity than our GP representation, and as such it is better to view the coefficients B as cell-averages. As such, we define $X_q \in \mathbb{R}^{d \times 2}$ as the locations of Gauss-Legendre quadrature points for all mesh cells (here we use a quadrature rule of order two). Then we define the highly sparse matrix $M \in \mathbb{R}^{|C| \times d}$, where $|C|$ such that the rows of $M$ correspond to individual

DG0 elements (of which there are $|C|$), and which contain for each column the associated quadrature weights normalized by the cell area. The resulting map from basis function coefficients $\mathbf{z}$ to finite element coefficients B is

$$\mathsf{B} = M \left[ L_q(C\mathbf{z} + \bar{\mathbf{w}}) + H_q \bar{\boldsymbol{\alpha}} \right] \tag{28}$$

The mean and marginal standard deviation of the resulting data-constrained distribution over $B$ is shown in Fig. 3.

### 3.2 Basal traction

Next we develop a similar representation for the basal traction field. This representation possesses many similarities to that of the previous section – we use the same low-rank Gaussian process as rendered tractable via structured kernel interpolation. However the traction is also a function of time, so we model it as a spatiotemporal Gaussian process that evaluated at discrete points in space $X$ and time $T \in \mathbb{R}^m$. The resulting $\beta(X,T) \in \mathbb{R}^{n \times m}$ is a random matrix with entries normally distributed as

$$P(\text{vec}\,\beta(X,T)) = \mathcal{N}(\boldsymbol{\mu}, K_t \otimes K_\mathbf{x}), \tag{29}$$

where $K_t \in \mathbb{R}^{m \times m}$ is a covariance matrix in the time dimension given by a covariance function and $\text{vec}$ is the vectorization operator (e.g. Magnus and Neudecker, 2019), which stacks the columns of a matrix into a vector. Here we take the mean function to be a learnable constant. Note that in writing the spatiotemporal covariance as a Kronecker product of temporal and spatial parts, we have made the common assumption of kernel separability, i.e. that variability in the space and time dimensions are *a prior* independent.

Following Sec. 3.1, we reparameterize $\beta(X,T)$ in terms of a finite basis. Using the properties of the Kronecker product, we can write

$$
\begin{aligned}
K_t \otimes K_\mathbf{x} &= (L_t L_t^T) \otimes (L_\mathbf{x} L_\mathbf{x}^T) \\
&= (L_t \otimes L_\mathbf{x})(L_t^T \otimes L_\mathbf{x}^T),
\end{aligned}
\tag{30}
$$



where $L_t \in \mathbb{R}^{m \times r_t}$ and $L_\mathbf{x} \in \mathbb{R}^{n \times r_x}$ are low-rank factors of their corresponding covariance matrices. This immediately leads

to the whitened linear model

$$\text{vec}\,\beta(X,T) = (L_t \otimes L_\mathbf{x})\mathbf{z} + \mu$$
$$\mathbf{z} \sim \mathcal{N}(0,I), \tag{31}$$

with $\mathbf{z} \in \mathcal{R}^{m_t m_x}$, i.e. a block-vector with each length $m_x$ block containing a temporal snapshot of the spatially varying traction field. We can evaluate this efficiently without forming the Kronecker product of covariance matrices as

$$\beta(X,T) = L_\mathbf{x}\,\text{mat}(\mathbf{z}_\beta)\,L_t^T, \tag{32}$$

where $\text{mat}$ is the inverse of the $\text{vec}$ operator.

We construct $L_\mathbf{x}$ as described in the previous section, here using a Matern covariance function with characteristic length scale of 3km (where we assume that traction varies at the same length scale as the topography), characteristic amplitude of unity, and differentiability index of 3/2. We construct $L_t$ similarly, but with a squared exponential covariance and a correlation

scale of half a year. Because the temporal covariance matrix is one-dimensional, it is small and there is no need for structured kernel interpolation - we simply compute it as $L_t = U\Lambda^{\frac{1}{2}}U^T$, where $U$ and $\Lambda$ are respectively the eigenvectors and eigenvalues of $K_t$.

As in Sec. 3.1, we must also map to finite element coefficients. However, because SpecEIS defines traction using a CG1 finite element basis (which linearly interpolates between mesh nodes), we simply evaluate Eq. 32 at the locations of mesh

nodes,

$$\beta(T) = \beta(X_{node}, T). \tag{33}$$

This parameterization has 9,443 degrees of freedom per year.

### 3.3 Surface mass balance

We seek to follow the same general recipe in parameterizing the specific mass balance as above, but this is complicated because

specific mass balance $\dot{a}(\mathbf{x}, S(\mathbf{x}), t)$ is primarily a function of elevation and time, with map-plane variability due to a variety of complex effects such as rain shadowing and insolation. Rather than parameterize a full 4D GP, we simplify the problem by assuming that the horizontally varying component of the mass balance is static in time, while the elevation component varies

$$\dot{a}(X, S(X), T) = \dot{a}_\mathbf{x}(X) + \dot{a}_z(S(X), T). \tag{34}$$

We model the first term as a zero-mean Gaussian process with a squared exponential covariance function with length scale 25km

and amplitude of 0.3ma[1]). We decompose as before into a low-rank factor $L_\mathbf{x} \in \mathbb{R}^{n \times r_\mathbf{x}}$ which leads to the finite-dimensional model

$$\dot{a}_\mathbf{x}(X) = L_\mathbf{x}\mathbf{w}_\mathbf{x}$$
$$\mathbf{w}_\mathbf{x} \sim \mathcal{N}(0,I). \tag{35}$$





We define the second term – which again describes the dependence on elevation and time – as

$$\dot{a}_z(S(X), T) = L_z[c_1 \mathbf{w}_z + c_2 \operatorname{mat}(\mathbf{w}_t) L_t^T]. \tag{36}$$

Here $L_z \in \mathbb{R}^{n \times r_z}$ is a piece-wise linear 'hat' functions evaluated at $S(X)$ (we note that we keep this elevation static and fixed to the surface elevation values given by the GLO-30 digital elevation model) with unit maxima defined at specified locations (or knots in the language of splines – the scheme here is equivalent to a spline of order 1). This implies piece-wise linear interpolation between specified elevations, namely sea level ($z = 0$m); the ELA in 2023 ($z = 900$m); the median elevation of the accumulation zone ($z = 1600$m); and the top of Mt. Logan ($z = 5950$m). $c_1 = 10 \mathrm{ma}^{-1}$ and $c_2 = 1 \mathrm{ma}^{-2}$ are characteristic scales of variability in surface mass balance rate and trend, respectively.

We assume that the SMB at each elevation changes as a linear function of time plus seasonal noise, which is to say that $L_t \in \mathbb{R}^{n \times (2+r_t)}$ is a scaled degree one Vandermonde matrix augmented with a low-rank representation of a temporal Gaussian process with a time scale of half a year. As such, in this work we do not explicitly parameterize specific surface mass balance as a function of external climate forcing, but rather attempt to infer it from various geometric observations. This simplification reflects a strong inductive bias, but is motivated by an exploratory analysis in which we attempted to parameterize SMB as a function of temperature and precipitation extracted from a variety of reanalysis products including MERRA-2, ERA-5, 20th Century Reanalysis V3, as well as direct observations from a weather station in nearby Yakutat, AK – All such products exhibit substantial disagreement with one another over our study area and typically do not extend over a sufficient time period to account for the entire historical modeling period from 1915 to 2023. Additionally, these modelling products are of insufficient resolution to account for the presence of the extreme topography characteristic of the St. Elias mountains – a well known challenge (Bieniek et al., 2016). Taken together, we find that the compounding errors associated with using such products overwhelms their utility compared to the simple parameterizations used here. The resulting model has 37 degrees of freedom.

### 3.3.1 Conditioning on SMB observations

We make use of a small observational data set of specific surface mass balance. In May 2023 we collected four snow cores at three locations in the accumulation area of Seward Glacier using a Kovacs drill to depths of approximately 7.5 meters, representing snow accumulation for the winter of 2022/2023 and total specific mass balance for 2021/2022 (Fig. 2a). For the 2022/2023 accumulation season, we measured 4.5 m of snowfall at an average density of 450 kg m$^{-3}$. For the 2021/2022 snowpack, we measured a total snow thickness of 2.9 m at an average density of 490 km m$^{-3}$, which corresponds to approximately 1.55 m a$^{-1}$ of ice equivalent. These observations are in rough agreement with observations from Sharp (1951), who estimated annual specific balance in ice equivalent of between 0.8 and 1.9 m a$^{-1}$ between 1945 and 1949 (approximately 12 km to the east at a similar elevation), and those collected by Marcus and Ragle (1970), who measured 2 m a$^{-1}$ of ice equivalent accumulation (collected prior to the ablation season) in 1965 (approximately 2 km to the northeast and at a similar elevation). In order to place these observations in a broader (although still limited) spatial context, we correlated these measurements with aerial radar (Li et al., 2019) measurements of snow accumulation collected at the end of spring in both 2018 and 2021. To do this, we applied a constant offset to the snow radar thickness measurements (separately by year) so as to match the 1.55 m





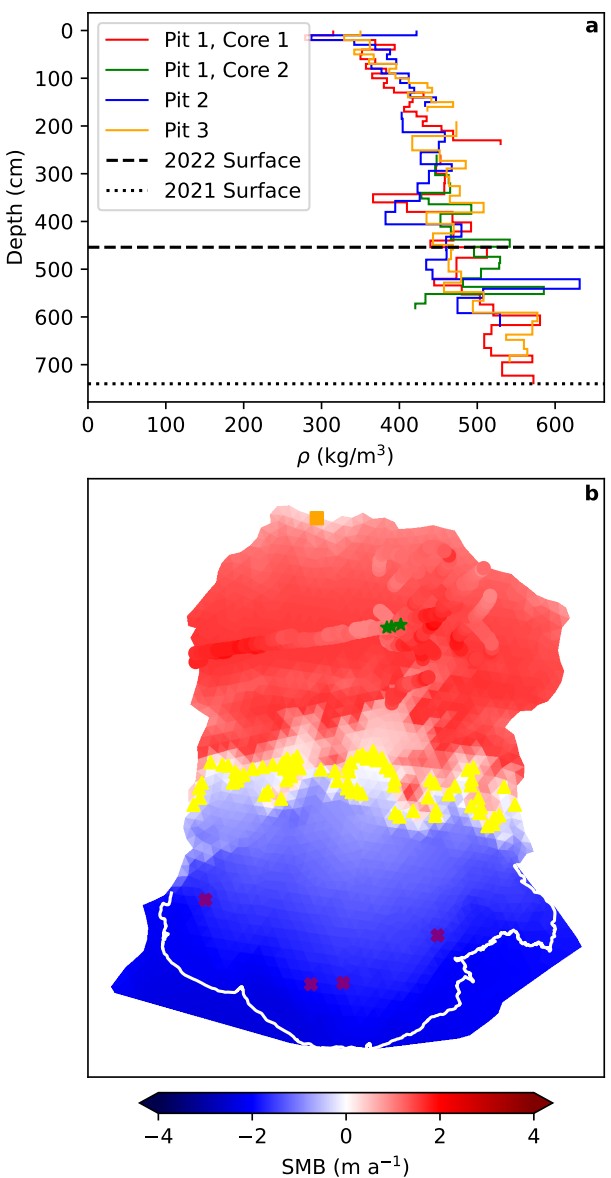

**Figure 2.** (a) Observations of surface mass balance from snow pits and cores, and (b) data-constrained prior mean for surface mass balance, with the orange square representing the Mt. Logan snow core, green stars the core measurements described in (a), yellow triangles the ELA, and purple crosses observed melt measurements.





$a^{-1}$ core observation from the 2021/2022 season (which represents net surface mass balance) at the point in the snow radar observations that is closest to the location of the core. We interpret the resulting product as representing both accumulation and ablation.

We infer the approximate position of the equilibrium line altitude from Landsat-8 images taken in September of 2023, which provides an implicit observation for locations with an annual surface mass balance of zero – however, there is significant uncertainty in this assessment. As a final constraint, we utilize a core taken from near the summit of Mount Logan and described in Moore et al. (2002), which shows an approximate and relatively stable high-elevation annual mass balance rate of 40 cm $a^{-1}$ (with a small increasing trend). While we have some limited data on ablation rates at various locations on the Sít' Tlein lobe;

we do not explicitly use these for model calibration, instead retaining them for validation of inferred melt rates as described below.

We also introduce a pseudo-observation corresponding to the glaciological steady-state condition

$$\int_{\Omega_{2013}} \dot{a}\, \mathrm{d}\Omega = 0, \tag{37}$$

where $\Omega_{2013}$ is the spatial extent of ice circa 2013. We only apply this pseudo-observation to the non-time-varying component

of the surface mass balance parameterization – this has the effect of producing a more (numerically) well-behaved prior that postulates that the true time-varying mass balance field is the result of a perturbation to one that would have yielded the 2013 ice extent, but does not actually restrict the space of admissible solutions.

We model each of these observations similarly, with

$$P(\hat{a}|\dot{a}(X_{obs}, S(X_{obs}), T_{obs})) = \mathcal{N}(\dot{a}(X_{obs}, S(X_{obs}), T_{obs}), \Sigma_{obs}), \tag{38}$$

with $\Sigma_{obs}$ a diagonal matrix populated with the observation variance for the modality associated with that observation. We assume (stated here in terms of standard deviation) this uncertainty to be 0.25m $a^{-1}$ for both the snow core/radar and ELA observations (an approximate scale of inter-annual variability based on Sharp (1951)); and 0.05ma$^{-1}$ for the ice core, which exhibits much less temporal variability based on Moore et al. (2002).

Defining $\mathsf{L}_{obs} = \begin{bmatrix} \mathbf{1} \otimes L_{\mathbf{x},obs} & \mathbf{1} \otimes c_1 L_{z,obs} & L_t \otimes c_2 L_{z,obs} \end{bmatrix}$, with $\mathbf{1}$ a column vector of ones with length $m$, we have that

$\mathrm{vec}\dot{a}(X_{obs}, S(X_{obs}), T_{obs}) = \mathsf{L}_{obs}\,\boldsymbol{\omega}, \tag{39}$

where

$$\boldsymbol{\omega} = \begin{bmatrix} \mathbf{w_x} \\ \mathbf{w}_z \\ \mathbf{w}_t \end{bmatrix}. \tag{40}$$

The above evaluates all locations and times at which an observation exists, but observations may not necessarily exist at all times. As such, we also define an observation operator $O$, which is a sparse matrix with ones corresponding to the location and 395 time where there actually exists an observation in $\hat{a}$.



The resulting solution to the least squares problem is given by

$$P(\boldsymbol{\omega}|\hat{a}) = \mathcal{N}(\bar{\boldsymbol{\omega}}, \Sigma)$$
$$\Sigma = \left[I + F^T \Sigma_{obs}^{-1} F\right]^{-1}$$
$$\bar{\boldsymbol{\omega}} = \Sigma F^T \Sigma_{obs}^{-1} \hat{a}, \tag{41}$$

with $F = O\,L$, and where we have assumed a prior mean of zero. It is straightforward to then compute a matrix root $CC^T = \Sigma$ such that the observation-constrained distribution over the mass balance field is given by

$$\mathrm{vec}\dot{a}(X, S(X), T) = \mathsf{L}\left(C\mathbf{z} + \bar{\boldsymbol{\omega}}\right)$$
$$\mathbf{z} \sim \mathcal{N}(0, I). \tag{42}$$

The surface mass balance in SpecEIS is represented in the DG0 basis, so mapping to the finite element basis coefficients a is performed identically to Eq. 28.

### 3.4 Initial Thickness

The last parameter that we must define a distribution over is $H_0(\mathbf{x})$, the initial thickness for the simulation (usually defined at some arbitrary time). It is challenging to develop a tractable representation in a similar manner to those described above because it is the only one that also corresponds to a physical quantity that is predicted by the flow model. It stands to reason that the initial simulation thickness ought to be one that is consistent with – or could have been generated by – the model itself. If this is not the case, then any prognostic model runs beginning from this initial state must attribute some of its dynamical behavior to the shape of the ice surface changing to be consistent with the flow model's physics as well as the structural assumptions of the other parameters as described in the previous three sections. Such an observation is not new, and is the motivation for long (relative to the forecasting period) spin-ups of ice sheet models, in which a flow model is run to approximate steady-state, sometimes with modifications to bring this steady state geometry into closer alignment with observations.

Here we use a data-constrained spin-up to eliminate the initial condition from consideration as a parameter. Specifically, instead of parameterizing the initial thickness through some basis representation, we take it to be given by the approximate steady-state solution produced by integrating the ice dynamics model over 2500 years with geometry given by Eq. 27, traction given by the time-average of Eq. 32, and the surface mass balance given by Eq. 39 evaluated at some reference time that we take to be the start time for further time dependent simulations. For this reference time, we take $t = 1915a$, which is approximately when the maximum little ice age ice extent occurred at Sít' Tlein based on geologic evidence, early cartography, and local knowledge (Tarr and Martin, 1914; Sharp, 1958). We note that we do have to specify *an* initial guess for this steady-state-finding routine (because we do not have numerical methods than can directly solve for the steady state without pseudo-time stepping, as in Bueler (2016),) and for this we use the 2013 surface elevation reported in the GLO-30 DEM. However, barring hysteresis, this initial condition does not influence the final steady-state solution that we use as the initial condition for further simulations.





## 4 Joint inference of model parameters

We now turn to the simultaneous inference of all parameters conditioned on all data. Written in terms of the finite basis

function coefficients described above and using some conditional independence assumptions (namely that direct observations

of one parameter do not affect any other parameters *a priori*), we write the joint posterior distribution over the combined

coefficient vector $\boldsymbol{\zeta} = \begin{bmatrix} \mathbf{z}_B^T & \mathbf{z}_\beta^T & \mathbf{z}_{\dot{a}}^T \end{bmatrix}^T$ as

$$
P(\boldsymbol{\zeta}|\hat{S}, \hat{B}, \hat{a}, \hat{\mathbf{u}}) \propto
$$
$$
P(\hat{u}|\boldsymbol{\zeta}) \times P(\hat{S}|\boldsymbol{\zeta})
$$
$$
\times P(\mathbf{z}_B|\hat{B}) P(\mathbf{z}_\beta) P(\mathbf{z}_{\dot{a}}|\hat{a}), \tag{43}
$$

where – unlike in the previous sections – we have explicitly subscripted each set of parameter coefficients to disambiguate them

from one another. The prior distributions are described in the previous sections. In order to evaluate this function, it remains to

specify likelihood functions for spatio-temporal observations of the surface elevation and velocity.

### 4.1 Surface elevation observations

We utilize two types of surface elevation observations. First, we use the publicly available Copernicus GLO-30 Digital Ele-

vation Model. This is the only product that offers wall-to-wall coverage over Sít' Tlein, and has a nominal date of 2013 with

estimated accuracy of 10m. Second, we use elevations derived from airborne laser swath mapping, which were collected be-

tween 1995 and 2021 as either part of Operation IceBridge (Larsen et al., 2015) or an earlier campaign by Keith Echelmeyer

(Arendt et al., 2002). These data were collected opportunistically, and vary widely with respect to coverage, with earlier surveys

characterizing only a predefined 'centerline' and 'cross-section', whereas later surveys flew a denser grid-like pattern, espe-

cially over the piedmont lobe (See Fig. 6 for spatial coverage and elevation relative to the GLO-30 DEM for each observation

year). These products have nominal error of less than 2m.

For all elevation products we resample so as to have a density of approximately 1 observation per 500m$^2$, which we assume

to be a safe minimum distance for assuming uncorrelated measurement error. We assume that a surface observation $\hat{S}_{ij}$ at

location $\mathbf{x}_i$ and time $t_j$ is distributed as

$$
P(\hat{S}_{it}|S(\mathbf{x}_i, t)) = \mathcal{N}(S(\mathbf{x}_i, t), \sigma^2 I), \tag{44}
$$

and $S(\mathbf{x}_i, t)$ is the true (or predicted) surface elevation at same. We use for the observational standard deviation $\sigma_S = 25$ m,

which is significantly inflated relative to the nominal accuracy posted for each product in order to account for un-modeled

effects such as seasonal variability, firn densification, and inaccurate error characterization.

$S(\mathbf{x}, t)$ is produced by the finite element model, which expresses the surface elevation based on the piecewise constant DG0

basis. As before, this is problematic because the coefficients represent, in a physical sense, cell averages rather than the actual

surface. To partially circumvent this and to produce a smooth version of the surface, we model the surface elevation using the

same (unconstrained) basis as used for the bed and solve a small least squares problem to get basis coefficients (here taken to



be the same as those used to model the bed) that produce a best fit at a given time $t$ corresponding to observation $j$

$$P(\mathbf{z}_t|\mathsf{S}_t) = \mathcal{N}(\bar{\mathbf{z}}_t, \Sigma_t)$$
$$\Sigma_t = (ML_q)^T \sigma_{model}^{-2} M L_q + I)^{-1}$$
$$\bar{\mathbf{z}}_t = \Sigma_t (ML_B)^T \sigma_{model}^{-2} (\mathsf{S}_t - \mathbf{h}_B(X)\boldsymbol{\alpha}_B), \tag{45}$$

where $\sigma_{model}$ is the assumed model error (here taken to be 1m), and $\mathsf{S}_t = \mathsf{B} + \mathsf{H}_t$. The full likelihood is then given by

$$P(\hat{S}_{it}|\mathsf{S}_t) = \int P(\hat{S}_{it}|\mathbf{z}_t)P(\mathbf{z}_t|\mathsf{S}_t)\,\mathrm{d}z_t$$
$$= \mathcal{N}(\bar{S}_{it}, \Sigma_{it}'), \tag{46}$$

where

$$\bar{S}_{ij} = L(\mathbf{x}_i)\bar{\mathbf{z}}_t + \mathbf{h}_B(\mathbf{x}_i)\boldsymbol{\alpha}_B$$
$$\Sigma_t' = \underbrace{\sigma^2 I}_{\text{Observational err.}} + \underbrace{L(\mathbf{x}_i)\Sigma_t L(\mathbf{x}_i)^T}_{\text{Projection err.}}, \tag{47}$$

and where we have used $L(\mathbf{x}_i)$ to represent the topography basis evaluated at observation point $\mathbf{x}_i$. The covariance matrix now represents two sources of error; first, the irreducible observational uncertainty, and second, the ambiguity in model-based surface elevation predictions due to the fact that the ice dynamics model's representation of the ice geometry is a *spatial average*. This also handles cases in which basis function coefficients to be inferred lie outside the finite element mesh, and thus cannot be informed by the model. While $\Sigma_S'$ is no longer diagonal, the second term is low-rank so the evaluation of its inverse (which is necessary for evaluation the likelihood) can be accomplished efficiently using an application of the matrix inversion lemma.

## 4.2 Surface velocity

We also constrain model parameters via observations of surface velocity. For this project we use an adapted and standardized version of ITS_LIVEv1 (Gardner et al., 2018), a worldwide velocity product derived through a speckle-tracking cross-correlation method applied to LandSats 5,6,7, and 8. We use annual velocity mosaics, which have 120m resolution and are available from 1985 until 2019. The nominal error in ITS_LIVE is variable, but on the order of 20 m a$^{-1}$. At Sít' Tlein, ITS_LIVE does not always have full coverage, particularly in the earlier years. As before, we downscale the observational density to one per 500m × 500m. We assume that each component of the velocity is normally distributed around the true (or predicted) value

$$P(\hat{\mathbf{u}}_{it}|\mathbf{u}(\mathbf{x}_i, t)) = \mathcal{N}(\mathbf{u}(\mathbf{x}_i, t), \Sigma_u) \tag{48}$$

with $\hat{\mathbf{u}}_{it}$ ITS_LIVE velocity vector observed at location $\mathbf{x}_i$ and time $t$, and $\mathbf{u}(\mathbf{x}_i, t)$ the modeled surface velocity at time $t$ evaluated at the observation locations. We assume an observational standard deviation of 50ma$^{-1}$ – however the error statistics





of ITS_LIVE are not well-understood and this number is somewhat arbitrary. Contrary to the previous section, we directly evaluate the predicted surface velocity from the finite element representation, as this representation is (almost) continuous and because we cannot come up with a more suitable representation beyond that of the model.

### 4.3 Evaluation of the log-posterior

It is more numerically convenient to work with the logarithm of the posterior distribution, which we call $\mathcal{J}(z_B, Z_\beta, z_{\dot{a}})$

$$
\mathcal{J}(\boldsymbol{\zeta}) = \mathcal{L}(\boldsymbol{\zeta}) + \mathcal{I}(\boldsymbol{\zeta}) + C
$$
$$
\mathcal{L}(\boldsymbol{\zeta}) = \sum_{(i,t) \in \mathcal{D}_\mathbf{u}} \log P(\hat{u}_{it} | \boldsymbol{\zeta})
$$
$$
+ \sum_{(i,t) \in \mathcal{D}_S} \log P(\hat{S}_{it} | \boldsymbol{\zeta})
$$
$$
\mathcal{I}(\boldsymbol{\zeta}) = \boldsymbol{\zeta}^T \boldsymbol{\zeta} \tag{49}
$$

where $\mathcal{L}$ is the log-likelihood with respect to the (yet-unused) observations of surface elevation and surface velocity; and $\mathcal{I}$ is the log-prior distribution, which is exceptionally simple on account of our chosen reparameterizations for each of these quantities, which renders all parameters uncorrelated and standard Gaussian. The summations in the above are taken over the sets of spatiotemporal points $\mathcal{D}_\mathbf{u}$ and $\mathcal{D}_S$ where observations exists for the given modality. We also note the presence of the constant $C$, which is a constant corresponding to the denominator in Bayes' rule that does not depend on the parameters.

While Eq. 49 describes our desired probability in formal terms, it is also helpful to describe its evaluation narratively. Given values of $\mathbf{z}_B$, $\mathbf{z}_\beta$, and $\mathbf{z}_{\dot{a}}$ (which initially have mean zero but which might be modified either through optimization or sampling), we map these to finite element model parameters using our various constructed bases (and with the traction averaged over time and the mass balance evaluated c. 1915 as described in Sec.3.3). We then use SpecEIS to compute a steady state which we take to be representative of the ice geometry in 1915. Using this geometry as an 'initial state' of sorts, we run the model forward in

time using a static bed elevation and time-varying traction and surface mass balance. Upon reaching the year 1985 (i.e. after 70 years of simulation time), observations of velocity and/or surface elevation become available, and while continuing to integrate the model forward in time until 2019, we also accumulate commensurate log-likelihood terms. At the end of the period of observations we also add the log-prior's contribution to the log-posterior.

The above computations are performed with Pytorch, which builds a computational graph of all operations, including the

SpecEIS function described in Sec. 2. As such, after computing $\mathcal{J}$, we can perform reverse-mode automatic differentiation on this graph, which computes the gradient of $\mathcal{J}$ with respect to every intermediate computation in the graph. Importantly, this gradient computation also propagates through the pseudo-timestepping of the initial steady-state computation, which implies that the influence of the parameters on this initial condition – and its resulting teleconnection with the posterior log-probability – is accounted for.



### 4.4 Computation of the Maximum A Posteriori point

With a well-defined maximization problem in hand along with the gradients of log-posterior, we can employ a gradient-based optimization scheme to find the most probable values for $\mathbf{z}_B$, $\mathbf{z}_\beta$, and $\mathbf{z}_{\dot{a}}$. The problem is unconstrained, but (empirically) exhibits strong correlations between parameter values and is high-dimensional. As such, we employ the classic quasi-Newton algorithm L_BFGS with line search (Zhu et al., 1997). We use a relatively short memory of 20 iterations, which ensures that parameter updates that occur in the first few iterations - which can be large and in directions inconsistent with later iterations - do not impede later fine tuning. We find that the algorithm so configured finds an optimum in a few hundred iterations (noting that each of these iterations is relatively expensive, involving upwards of 120 forward and adjoint solutions per likelihood evaluation).

### 4.5 Steady-state problem for initial guess and estimation of model bias

Here we outline one additional trick that we use to improve our results. Prior to the solution of the full, time-dependent minimization problem, we solve a reduced inference problem in which we find the maximum a posteriori (MAP) solution to Eq. 49 but for only a single (with respect to time) surface elevation (the GLO-30 DEM) and a single (with respect to time) velocity field (The ITS_LIVE average mosaic over the 34 year data record). We adopt steady state dynamics in which we only use time-averages of the parameters, and run the flow model to a steady state for comparison with observations. The resulting solution serves two purposes; first, it provides an initial guess for the full time-dependent problem that is already very close to optimal, particularly for $\mathbf{z}_B$. Second, because the steady components of the parameters are much fewer than the full time-varying fields (and because this solution is insensitive to the initial surface elevation $S_0$), the system is less likely to overfit the observations. As such we interpret the resulting residuals between model predictions and observations as an irreducible bias resulting from model mis-specification. In subsequent calculations, we subtract this bias from model predictions, which limits the potential for other parameters associated with time-varying fields from compensating for this bias in non-physical ways. As an example, foregoing this correction can lead to a tendency for the optimizer to make the contemporary precipitation fields too small in an effort to match the zero velocity in ice free areas evident in ITS_LIVE. Zero velocity is, of course, not physical – the annual accumulation in these areas is positive and the balance of fluxes requires that the annual-average velocity in such places should be quite high. However, because the primary mechanism for mass transport is avalanching down to bare earth, ITS_LIVE cannot account for such processes (whereas the model does account for mass transport from these regions). By adopting the bias correction approach described above, we largely circumvent this issue.

### 4.6 Approximation of the posterior covariance

We seek to approximate the complete posterior given by Eq. 43, yet the procedure above yields only the most probable set of parameter values with respect to the posterior distribution (the so-called maximum *a posteriori* or MAP point). To quantify the posterior uncertainty, we employ the Laplace approaximation, which approximates the posterior distribution as a multivariate normal with the MAP point as its mean. The covariance matrix is then determined through a second-order Taylor expansion





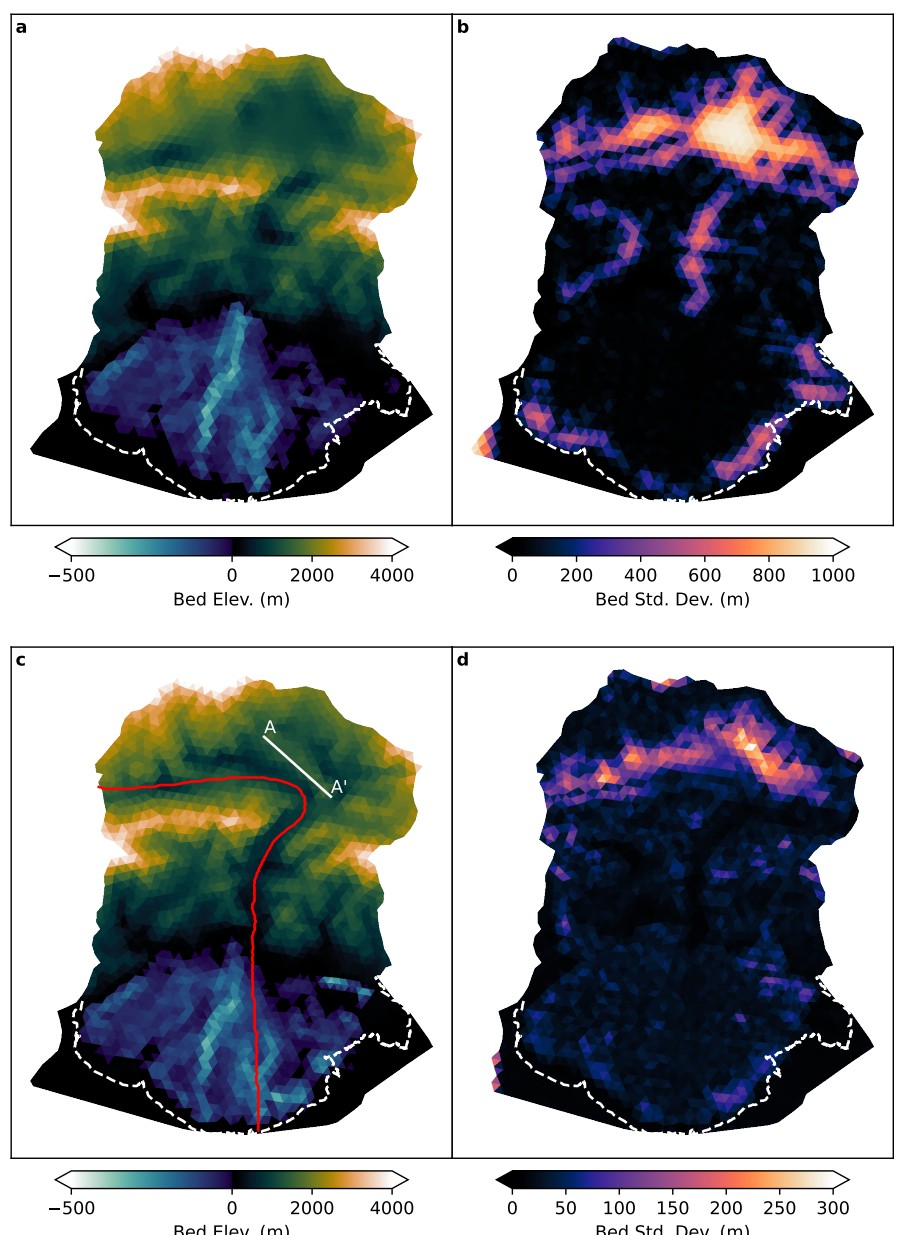

**Figure 3.** The prior mean (a), prior marginal standard deviation (b), posterior mean (c) and posterior marginal standard deviation (d) of the inferred bed elevation. The red line corresponds to the profile plotted in Fig. 4





of the log-posterior. Note that we do not characterize posterior uncertainty in the time varying component of basal traction, which is very high-dimensional and requires prohibitive computation to characterize; however, we do characterize the posterior covariance over the static traction field.

Expanding the log posterior around the MAP point we have that

$$
\begin{aligned}
\mathcal{J}(\boldsymbol{\zeta}) \approx{} & \mathcal{J}(\boldsymbol{\zeta}_{MAP}) \\
& + \frac{\partial J}{\partial \boldsymbol{\zeta}}(\boldsymbol{\zeta} - \boldsymbol{\zeta}_{MAP}) \\
& + \frac{1}{2}(\boldsymbol{\zeta} - \boldsymbol{\zeta}_{MAP})^T \mathcal{H}(\boldsymbol{\zeta} - \boldsymbol{\zeta}_{MAP}).
\end{aligned}
\tag{50}
$$

By the definition of the MAP point, the first-order term is zero, and we recognize this approximated log-posterior as a normal

distribution with a covariance matrix given by the negative of the inverse of the Hessian $\mathcal{H}$. Following Bui-Thanh et al. (2013) and Isaac et al. (2015), we decompose this Hessian into prior and likelihood parts

$$
\mathcal{H} = (\mathcal{H}_{data} + I).
\tag{51}
$$

The Hessian of the prior (which, once again, is unit-Gaussian and uncorrelated) is the identity matrix, while $\mathcal{H}_{data}$ is the Hessian of $\mathcal{L}$ with respect to $\boldsymbol{\zeta}$. Direct computation of the data Hessian is intractable, both because it is very large and as such would be difficult to form - let alone invert - and also because it would require $m$ function evaluations to compute. We instead

approximate the inverse Hessian using a randomized matrix decomposition as described in Appendix B.

## 5 Probabilistic forecasting

In the previous sections we developed a method for quantifying an approximate set of parameter values that produces model predictions that are consistent with observations, i.e. the posterior distribution over parameters corresponding to the second

term inside the integrand in Eq. 52. We now turn to using these parameter values to make predictions about future change at Sít' Tlein. Our approach to this problem is simple and does not require a complex numerical treatment – we take the classic ensemble modelling approach of drawing as large a set of parameters as computationally feasible and run the model forward in time with those parameters. The resulting approximation to the predictive distribution is

$$
\begin{aligned}
P(\Delta(t)|\mathbf{d}, \mathcal{F}) &\approx \frac{1}{n_s} \sum_{j=1}^{n_s} P(\Delta(t)|\mathbf{m}_j, f_j) \\
\mathbf{m}_j &\sim P(\mathbf{m}|\mathbf{d}) \\
f_j &\sim P(f|F).
\end{aligned}
\tag{52}
$$


While such Monte Carlo methods converge slowly, they are easy to implement and are embarrassingly parallel.

We integrate the model from a steady state at 1915 to 2344. Each model has a different resulting geometry and pattern of flow, and from these we evaluate quantities of interest at different times. We qualitatively divide the predictive distribution into a



hindcast (from 1915-2023) and forecast (from 2023 onward). The distribution over parameters for the hindcast is unambiguous, but forecasting requires some assumptions about future changes to time-evolving parameters.

**Steady geometry** We assume that the bed elevation remains constant (although we relax this assumption for a later experiment).

**Periodic surges** We assume that the 36 year inferred record of time-variation in the basal traction repeats in a periodic fashion.

While we do not believe that this approach will necessarily predict the precise location, timing, and magnitude of future surges, we believe that it will capture their statistical features – and their resulting influence on geometric change. As is evident from the repeating nature of Sít' Tlein's looped moraines (Muskett et al., 2008), at least in the short term we do not expect surge dynamics to depart qualitatively from the pattern observed before, although this assumption is likely to become less valid if Sít' Tlein undergoes significant geometric change. Nonetheless, this choice is also governed

by necessity, since we currently lack a validated and general mechanistic model for sliding generally and one that can predict surging specifically.

**Projected and frozen mass balance** We explore two scenarios for future evolution of the surface mass balance. Recalling that we parameterize the surface mass balance for different elevations as a linear function of time since 1915, in our first scenario we linearly extrapolate these trends into the future, which would be roughly commensurate with a continued

linear increase in mean air temperatures. As a second scenario, we consider an end member case in which we freeze the SMB field at 2023 and hold it constant into the future, which corresponds to an immediate cessation of warming.

**Calving** One final consideration that we need to address is calving. While contemporary Sít' Tlein does not undergo significant mass loss due to calving, it does have small calving fronts on the margins of two proglacial lakes, and at least one of those lakes is already receiving significant tidal inputs of marine saltwater (Thompson et al., 2021). We therefore expect

that if the glacier undergoes additional retreat, it could develop a broad marine or lake-terminating calving front, as is the case for its nearby neighbors in Disenchantment Bay, Icy Bay, and the Bering Glacier. Because we cannot yet observe significant calving here, we cannot infer the values of parameters governing a forward calving model. While marginalization over a calving velocity prior is possible, here we simply examine two end-member scenarios to bracket possible future behavior. In the first, we assume that calving does not occur; this is not to say that ice does not float - we

do allow floating tongues to form. This is perhaps not unrealistic for lake terminating glaciers, which in coastal Alaska have been observed to develop significant floating tongues (Truffer and Motyka, 2016). In the second scenario, we adopt a calving-on-flotation criterion, and associate with floating ice a calving velocity of $1\mathrm{km\,a}^{-1}$ which roughly corresponds to the observed retreat rate at Columbia Glacier.

We perform ensemble experiments using combinations of each calving and climate evolution assumption stated above, for a

total of four experiments. In each of these four experiments, we assume both steady geometry and periodic surging.

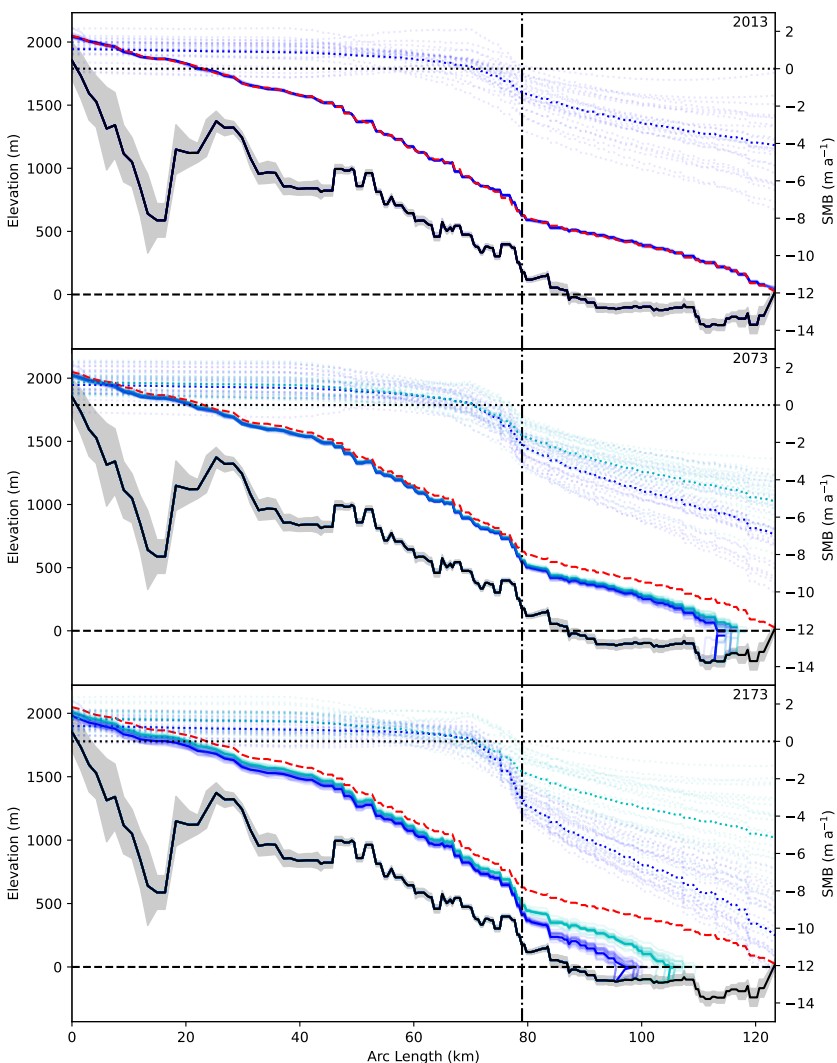

**Figure 4.** Profile of ice geometry corresponding to the red line in Fig. 3 at 2013 (a), 2073 (b), and 2173 (c). Cyan and blue solid lines are surface elevations corresponding to individual ensemble members for climate assumptions in which surface mass balance is frozen at 2023 values or extrapolated into the future, respectively. Cyan and blue dotted lines are the corresponding surface mass balance curves. The red dashed line shows the 2013 ice surface for reference. The black line indicates the most probable bed elevation, with the gray envelope showing two standard deviations. The vertical dashed and dotted line shows the approximate location at which the Sít' Tlein piedmont lobe begins.





# 6 Results

In this section we describe both the posterior distribution over data-informed model parameters alongside the predictive distributions over both the hindcast and forecast periods generated by sampling from the posterior and running the ice dynamics model from 1915 until 2344. An analysis of model performance against unseen data is given in Appendix C.

## 6.1 Bed geometry

We begin with an analysis of the inferred bed elevation. Figure. 3c shows the most probable bed elevation. Much of this map reflects direct observations of the bed taken via radar sounding; however significant features that were not imaged by radar, particularly in the accumulation area at the northern end of the map, are also captured. Most salient of these is the presence of a subglacial mountain range (Transect A-A' in Fig. 3c) that continues trending southeast from the flanks of Mt. Logan, which divides the principal accumulation area for the Seward Lobe into two separate regions. The surface expression of this feature as visible on a digital elevation model is subtle - however on the ground this feature is conspicuous and associated with a large crevasse field and an increase in ice surface gradient. Interestingly, the uncertainty in bed elevation over this region (shown in Fig. 3d) is relatively low, indicating that the surface observations of velocity and especially surface elevation strongly constrain the bed in this region. In contrast, the basins to the northeast and southwest of this ridge, while fast flowing, have relatively homogeneous topography and low gradients, which leads to significantly greater uncertainty. Another region that exhibits high uncertainty in elevation are the areas very near the margin of Sít' Tlein's piedmont lobe. This uncertainty is the result of the lack of dynamics in this region - without significant flow, the observations of velocity (which are nearly zero) and geometry (which is uninformative in low slope regions) provides little information and the posterior variance is nearly the same as that of the prior. While there are few bed measurements within the fast-flowing trunk of Seward Glacier as it flows through the gap in the St. Elias range, the posterior uncertainty there is similar to that in regions constrained by direct bed observations, indicating that ice dynamics provides a strong constraint.

It is also instructive to observe the geometry in cross-section. Figure 4a shows the inferred bed elevation along the red profile indicated in Fig. 3c. We see here also the pattern of relatively high bed uncertainty in the accumulation zone, with low uncertainty in areas with fast flow or direct measurements. This figure also shows the resulting surface geometry for 50 random samples drawn from the posterior distribution and integrated through time, evaluated at the year 2013. We see a strong agreement between random surface samples and the observed surface elevation, especially relative to the vertical scale of the system, in which all simulations are atop one another at this scale. The surface mass balance curves that associated with each of these profiles shows considerable variability, primarily due to annual noise that cannot be reduced by temporally-limited observations and the influence of which has little direct influence on annual surface elevation or velocity changes. Nonetheless, we infer a mean contemporary surface mass balance at sea level, which was not part of the observational dataset, to be approximately $4 \, \text{ma}^{-1}$.



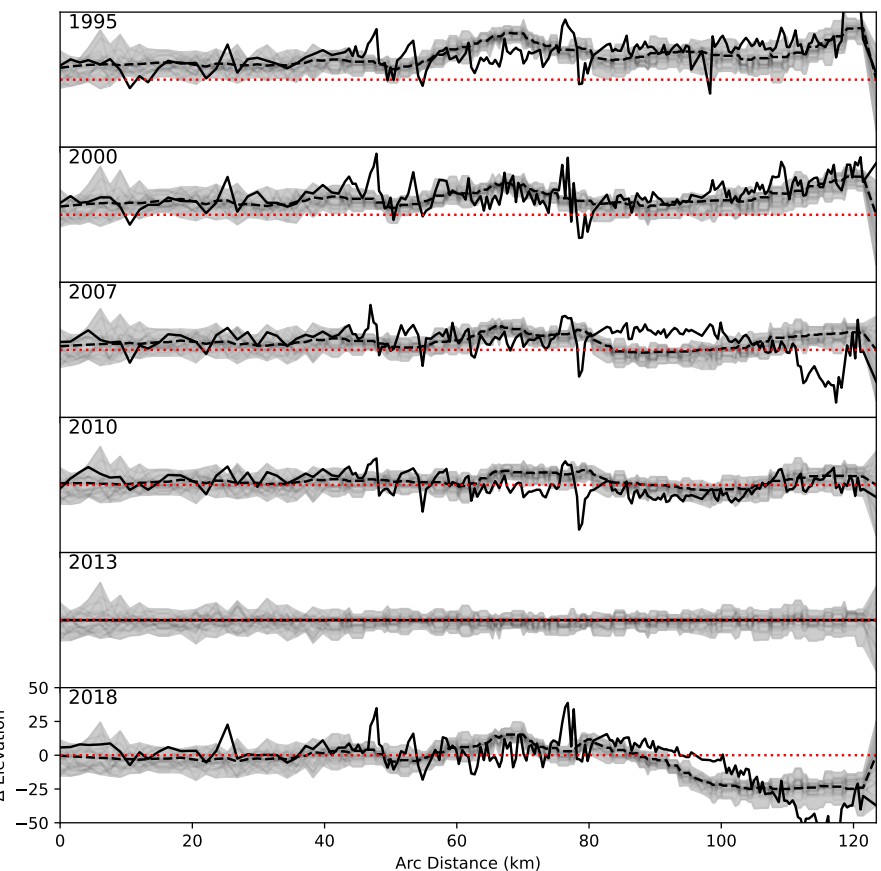

**Figure 5.** Observed (dashed) and MAP predicted (solid) surface elevations relative to the 2013 COP30 DEM for a selection of years plotted along the red line shown in Fig. 3. Envelopes represent the range of elevations for these computed for 50 random samples from the posterior distribution.

## 6.2 Elevation change

Zooming in on the surface and plotting the modelled and observed surface elevations relative to 2013 for years in which observations exist (Fig. 5), we find that the predicted surface elevation matches observations in both absolute magnitude and in

trend. However, there is still some spread in the distribution due to assumed uncertainties in the surface elevation observations. Figure 6 shows the spatial distribution of the model's most probable predicted elevation change relative to 2013 alongside sparse observations. We find good agreement between the broad spatial patterns, but the match is imperfect, particularly in later years over the piedmont lobe in which the data indicates a significant drawdown that is presumably the result of a surge that we have not adequately captured, alongside a perhaps too-simple surface mass balance parameterization. Of particular

scientific interest, it is evident from observations that the ablation zone is thinning much more quickly than is the accumulation zone, and the spatiotemporal variability in the inferred surface mass balance - and the resulting modeled thinning - reflects this



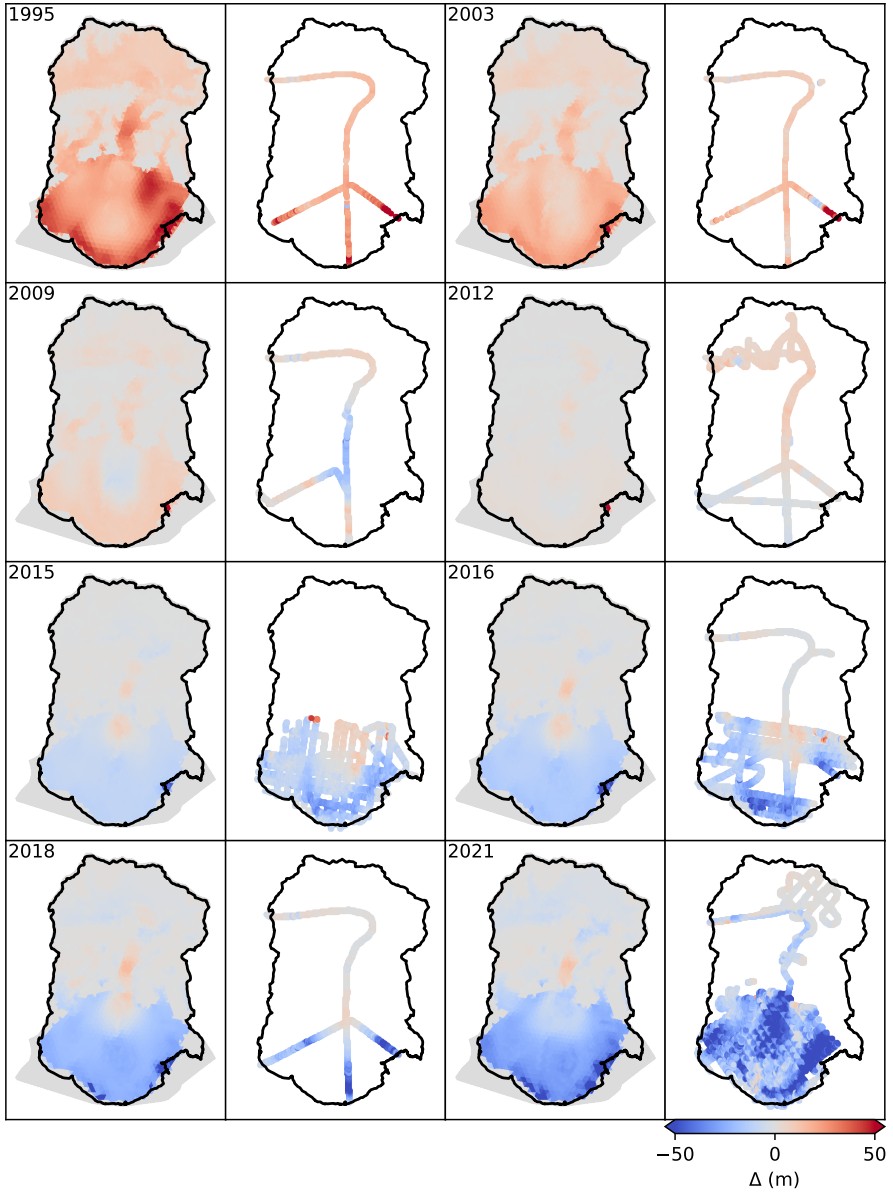

**Figure 6.** Modelled (left) and observed (right) surface elevations relative to 2013 for selected years.

pattern as well. The misfit between the model and observations is shown in Fig. 7. We generally find that the MAP surface approximates observations to within 20m over smooth, ice-covered regions.





**Figure 7.** Predicted surface minus observed surface for all years for which data is available. The color scale saturates at 20m.



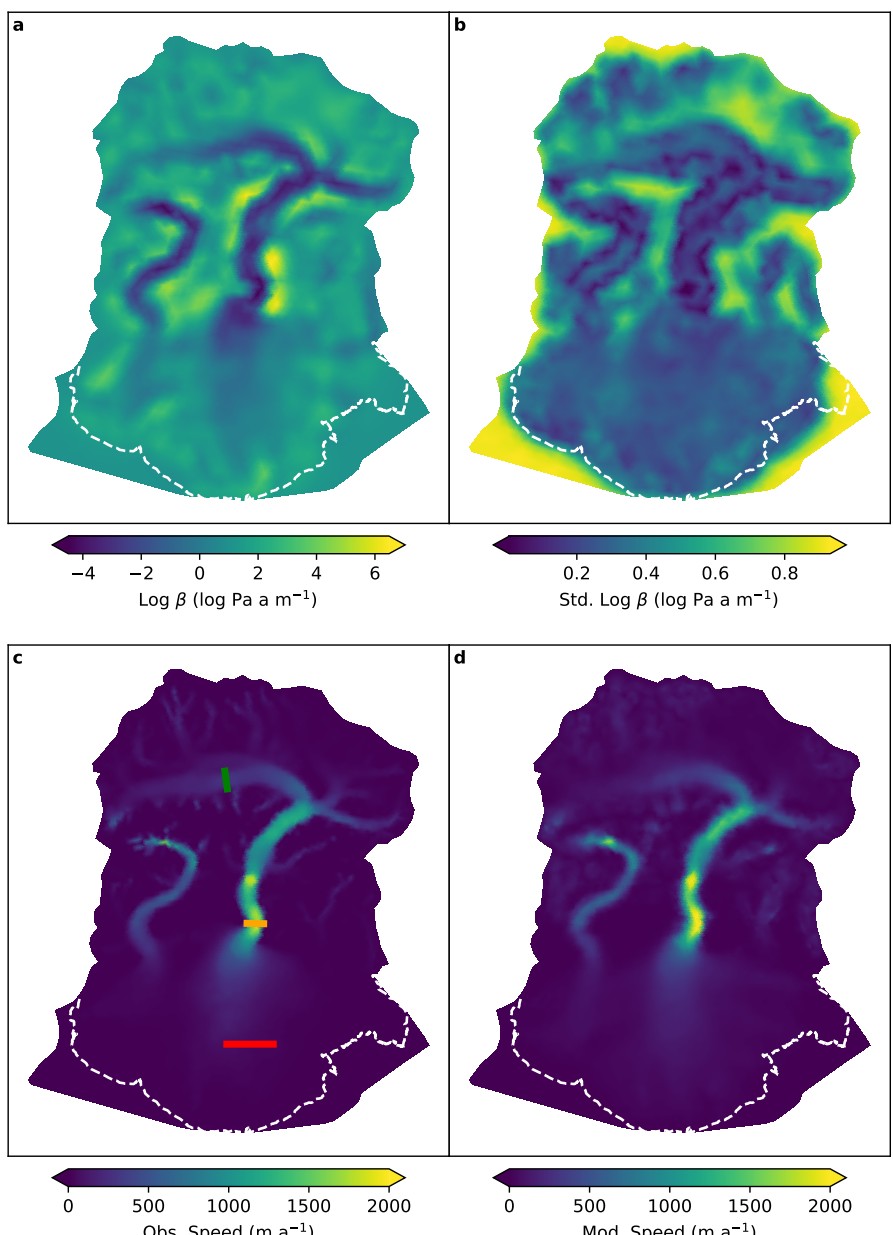

**Figure 8.** Mean (a) and standard deviation(b) of the basal traction field. Observed (c) and modelled (d) speed averaged over the years 1985-2019. Included in (c) are also transects over which velocity is averaged and displayed as a function of time in Fig. 9.





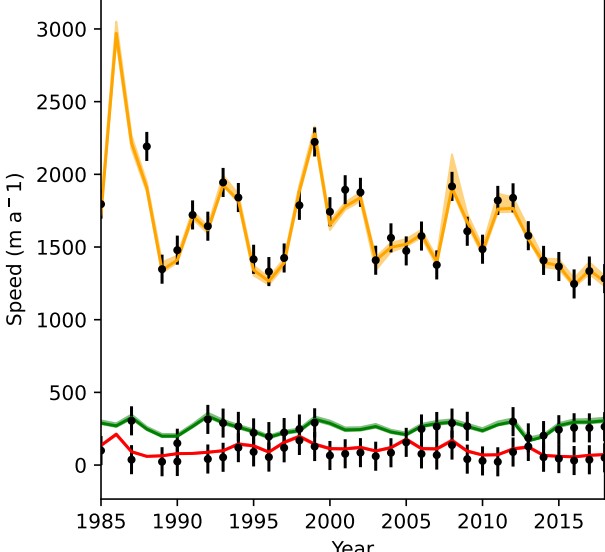

**Figure 9.** Time series of average speed computed over the equivalently colored cross-sections in Fig. 8c. Black points indicate ITS_LIVE annual velocity mosaics with assumed 2-$\sigma$ errors.

### 6.3 Traction and velocity

Fig. 8a shows the inferred mean basal traction field. regions of fast flow exhibit low traction, with relatively low posterior variance (Fig. 8b) in regions well-constrained by velocity observations. As is to be expected, steep or ice-free areas without velocity measurements exhibit a high posterior variance.

    More interesting than the traction fields themselves – which mostly alias unknown physical processes – is the resulting velocity field, the temporal mean of which is shown in Fig. 8d alongside the mean observation. While this is an expected result

since the inference of basal traction to match surface observations is well-established, we find good congruence between the modelled and observed value over areas with fast flowing ice. This fit is once again not perfect, particularly in steep regions (where the model predicts relatively fast flow that corresponds to what would likely be avalanche transport in the real world) and in slow regions of the lobe, where more flow than is identified in the observations is necessary to maintain contemporary geometries. This latter effect could be attributed to two (not mutually exclusive) possibilities. First, the flow rates in the

slow parts of the lobe may simply be below ITS_LIVE detection thresholds and so are spuriously assigned a zero velocity. Alternatively, it may be the case that some parts of the piedmont lobe are replenished by velocity configurations that do not exist in the 36 year observational record. Sít' Tlein is a surge-type glacier, so it is reasonable to imagine that some parts of the lobe were not recipients of upstream ice flux during this time period, yet the model must route ice to these areas to ensure that they are not modelled as ice-free. This in turn could be exacerbated by errors - particularly those induced by model inadequacy

- in the modelled surface mass balance field.



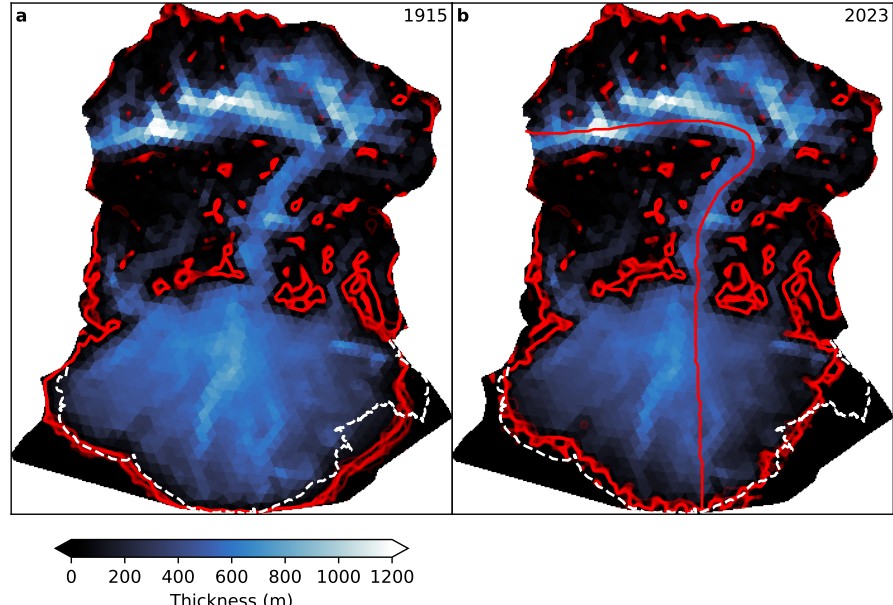

**Figure 10.** Predicted ice extent at the beginning and end of the hindcast period. Red lines indicate terminus positions of individual ensemble members. The white dashed line indicates the ice extent circa 2023.

While the primary long term dynamics of Sít' Tlein are likely controlled by time-averaged velocity, we also explicitly model a time-varying traction so as to match the evident surges in the observational record. Fig. 9 shows the time-series of velocity magnitude averaged over the three profiles shown in Fig. 8c. In Sít' Tlein's fast flowing trunk, we recover the time-series of velocity with high fidelity. Because the posterior uncertainty in traction is low there, and also because we do not capture
the posterior variance over the time-varying component of the traction, the predicted variance in velocity is also low. The moderately fast accumulation area exhibits similar properties. Again, we find that areas near the ice margin generally have flow speeds that are somewhat too fast, for reasons described previously. A complete spatially and temporally distributed comparison of predicted versus modelled velocity anomalies is shown in Figure S1.

### 6.4 Forecasted change

Figure 10 shows the evolution of ice thickness from 1915 to 2023. At the beginning of this hindcast period, we see Sít' Tlein in a relatively extended configuration, with a significant oceanic interface where the ice extended past the top of the contemporary foreland. While we did not use historic observations of early 20th century ice extent as a constraint, the model's inferred configuration in 1915 is in good qualitative agreement with maps from this time period (Russell, 1893; Tarr and Martin, 1914). In 2023, the ice extent and geometry match the present configuration by design. Figures 11 and 12 show the
ice extent and thickness of the piedmont lobe in 50 and 150 years under each combination of assumed calving and assumed future mass balance. Fifty years from present, we forecast with high probability that Sít' Tlein's lobe will disengage from the





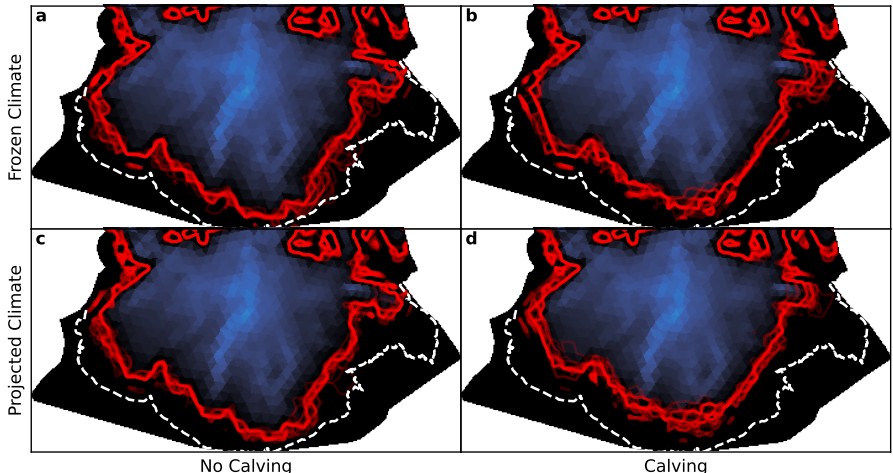

**Figure 11.** Predicted ice extent at 2073 under different assumptions of calving and projected surface mass balance. Red lines indicate terminus positions of individual ensemble members. The white dashed line indicates the ice extent circa 2023.

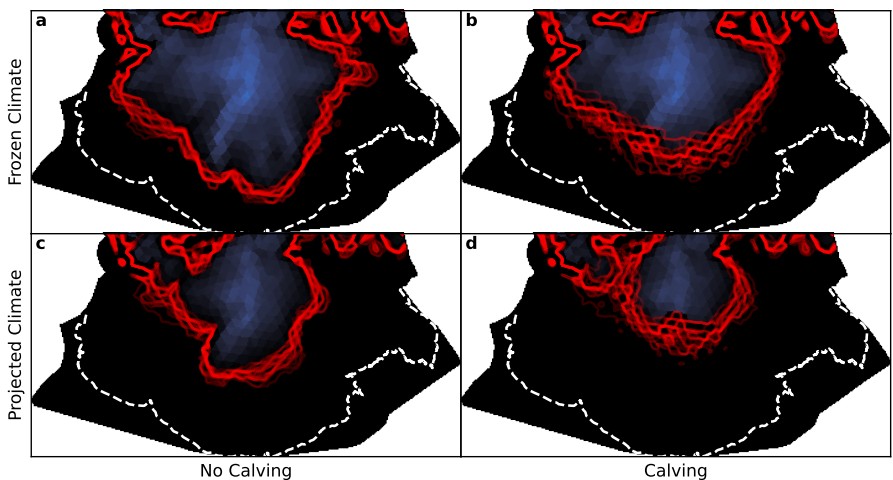

**Figure 12.** Predicted ice extent at 2173 under different assumptions of calving and projected surface mass balance. Red lines indicate terminus positions of individual ensemble members. The white dashed line indicates the ice extent circa 2023.




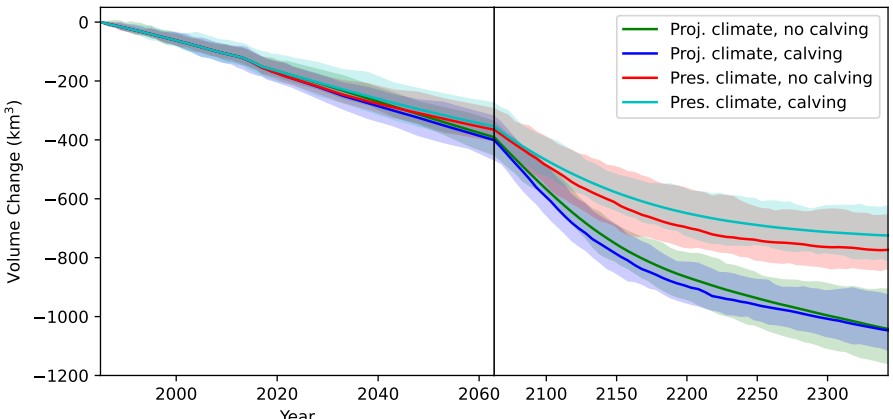

**Figure 13.** Volume evolution trajectories of Sít' Tlein from 1985 until 2344, with cyan and red corresponding to experiments with calving off or on, respectively, in which the surface mass balance is frozen at inferred 2023 values; and with green and blue corresponding to experiments with calving off or on, respectively, in which the surface mass balance is linearly extrapolated into the future. Shaded regions correspond to 95% credible intervals for each scenario. Note the change in axis spacing at the year 2073 to show detail over the historic period and the next 50 years.

foreland and will terminate in a lake or marine embayment of increasing size. This modelled configuration is qualitatively similar to the contemporary state of neighboring Bering Glacier, which may serve as a valuable analog (Lingle et al., 1993). The qualitative differences between scenarios are minimal, although the presence of calving and a warmer climate both lead to

slightly increased retreat. The differences in scenarios at 150 years from present are more dramatic; Under linearly projected warming, Sít' Tlein's piedmont lobe will likely have mostly disintegrated, which is exacerbated by calving. Under a frozen climate, the piedmont lobe persists, albeit with a volume reduction of between 80 and 90%, depending on calving dynamics. In the event that current forelands degrade in the absence of active glacier sedimentation, it could be the case that the Seward glacier will terminate in a shallow marine embayment, not dissimilar to neighboring Hubbard Glacier. However, the Sít' Tlein

complex's geometry is not conducive to further retreat up its tributary fjords, which have beds that are likely above sea level, a significant contrast to Hubbard Glacier.

    Figure 4b and c show forecasted geometry along profile along the red curve in Fig. 10b under both assumptions of linearly extrapolated and steady-at-2023 climate assumptions. As expected, the frozen climate exhibits less retreat, particularly at 2173 (also see Figs. 11 and 12). However, the qualitative description in the previous paragraph applies to both cases. Even if

warming does not increase beyond present values, Sít' Tlein will likely undergo a disintegration of the piedmont lobe by 2173, changing from a mostly terrestrial terminus to a dominantly lacustrine or marine one. Over the 50 year scale, we expect modest surface lowering in the accumulation area under any scenario. However, 150 years from present under continuing warming, we see nearly double the surface thinning compared to the frozen experiment. In both cases the magnitude of changes in the accumulation area are smaller and less certain than those in the ablation zone.





Finally, Fig. 13 provides the empirical distribution of volume change for each experiment as a function of time. The influence of calving is small for the linearly extrapolated climate - the surface mass balance forcing is so strong that calving plays only a transient role in determining the mass evolution of the system. For the frozen climate experiment, scenarios with calving enabled on average produce significantly greater mass loss. However, such simulations are also more uncertain, as calving interacts nonlinearly with different random bed topographies. For all scenarios, the mass loss at 50 years from present is

similar and between 300 and 600 gt. Because much of this ice is already below sea level, this translates into somewhat less than 1mm of sea level equivalent. At 300 years from present the variability is much greater, and the range of ice loss over all scenarios is between 700 and 1300 gt, corresponding to at least a complete loss of the piedmont lobe, and perhaps significant further mass loss in the erstwhile accumulation area.

## 7  Discussion

### 7.1  Significant model limitations

The model ensemble presented here represents our best effort to produce a credible prediction of future change at Sít' Tlein, including both a defensible representation of uncertainties and demonstrable skill at hindcasting. However, it still possesses significant limitations that should be considered when contextualizing these results.

### 7.1.1  Geometry models

First, our prior distribution over the bed elevation is simplistic relative to the true richness of the region's geomorphic reality. While convenient and flexible, a Gaussian process with fixed kernel parameters cannot capture the effects of the full range of geomorphological processes acting in this region, especially the significant qualitative difference between the topography that is currently beneath the ice (which is relatively muted and smooth) versus the subaerial topography comprising the St. Elias range's dramatic aretes and peaks (Cotton et al., 2014). In particular, our terrain model is in many places too smooth,

while in others too rough. Terrain models that rely on an adaptive basis, for example using generative adversarial networks trained on natural topography, might present an alternative (Voulgaris et al., 2021). However, a fundamental challenge remains in that the effective dimensionality required to represent spatial variability increases with decreasing spatial regularity, which is particularly challenging when trying to compute posterior covariance matrices. A similar criticism is reasonable for the basal traction, though we have little basis for understanding the spatial correlation structure for traction given that it is not directly

observable.

### 7.1.2  Spatial parameterization of mass balance

As emphasized previously, a critical limitation is our use of a simple parameterization of surface mass balance, which reflects our lack of detailed process understanding. This model is perhaps inadequate in several ways, particularly when taken in conjunction with our general lack of knowledge about the spatial distribution of accumulation and melt in the St. Elias range.





First, we assume SMB to vary with elevation according to a piecewise linear function with changepoints defined at sea level, the ELA, the median elevation of the principle accumulation zone, and at the highest elevation. This captures the most salient features of linear lapse rates for both temperature and precipitation (indeed, when combined with a degree day model, such assumptions typically yield a piecewise linear function for annually-averaged SMB, with a break in slope at the ELA), and also the effects of high-elevation precipitation rarification. Nonetheless, despite including the capacity for explicit spatial variation, we do not believe that this parameterization fully captures the region's dramatic orographic effects like rain-shadowing, topographic steering of precipitation, or the influences of solar aspect. Critically, this parameterization also does not take into account the influence of debris mantling, which could have a significant impact in the near-terminus region. Indeed, in some regions near the ice terminus, sufficient material has accumulated to support a rich plant community (including trees) and melt has essentially stopped. In such regions, the semantic distinction between glacial ice and ice cored moraine also becomes somewhat ambiguous. Regardless of such definitions, we lack validated models both for predicting debris thickness and for estimating its influence on melt rates.

### 7.1.3 Temporal parameterization of mass balance

Similarly, the variation of SMB in time is parameterized solely as a linear trend (and random interannual variability), which ignores potential knowledge of local and global temperature and precipitation trends – although the extrapolation of such knowledge at Sít' Tlein represents a significant challenge. Nonetheless, a better approach that could deal with both of these simplifications might be to use a high-resolution regional climate model (e.g. Bieniek et al., 2016) – perhaps in conjunction with linear orographic precipitation theory (Smith and Barstad, 2004) to accommodate additional topographic detail – to derive time-varying precipitation alongside a surface energy balance model (e.g. Hock and Holmgren, 2005) to estimate melt. However, such models have significant numbers of parameters themselves and are not easily amenable to integration within an automatic differentiation framework. Combined with the paucity of observations for the glacial system considered here, it is not clear that the resulting more sophisticated model would lead to an increase in predictive skill and high quality SMB modelling in mountains remains an active research area.

### 7.1.4 Ice-ocean interactions

Our model makes use of a simplified frontal ablation parameterization, and we explore two representative configurations of such physics, one in which frontal ablation does not occur, and one in which it occurs at a rate similar to a previously observed tidewater retreat. Nonetheless it is possible that this simplified approach could miss more nuanced processes. For example, our model does not explicitly account for the intrusion of warm, oceanic saltwater, which we know to be occurring at Sitkagi Lagoon and not at Malaspina Lake (yet, to our knowledge). As such, the details of potential tidewater retreat – particularly during early stages in which the glacier is still interacting significantly with its terminal moraine – might not be fully captured. Nonetheless, because the primary driver of the significant surface elevation change at Sít' Tlein is very likely to be a profound surface mass imbalance, we believe that these inaccuracies are unlikely to modify the longer term conclusions presented here regarding the stability of the piedmont lobe.



### 7.1.5 Models of observational uncertainty

Finally, with respect to the inference procedures described here, we make significant use of simplified models of measurement noise for all observed quantities. While we make heavy use of normal distributions to model noise in all of our datasets, it is very likely the case that all observations are biased in complicated (and unmodeled) ways, and also that the uncertainty characteristics admit outliers. We do not have a good understanding of observational uncertainty for most products even when provided. Even if we had a detailed understanding of marginal error statistics (i.e. the observational uncertainty in surface velocity at a single point), spatial and temporal correlations in error are unreported and unknown. While we have tried to be conservative in defining our likelihood models, it undoubtedly the case that we have induced additional model error through misspecification of such likelihoods.

### 7.2 The influence of basal sedimentation and alternate hypotheses for surface elevation change

A conspicuous feature of Sít' Tlein is its lack of significant mass loss through calving due to the presence of a large sedimentary morainal bank of varying composition (Thompson et al., 2024). As such, it is reasonable to imagine sedimentation playing a role in potential stabilization of the Sít' Tlein lobe as it undergoes tidewater retreat. To test this hypothesis, we perform a supplementary experiment in which we modify the basal topography such that the elevation of the bed at any location where it is both below sea-level and not in contact with the ice base (in other words the ice has retreated away from the location) is raised either to the ice base or to sea level. Thus this experiment represents a maximal sedimentary end-member where there is an infinite supply of sediment that is deposited instantly wherever possible. We apply this experiment to the present-climate case without calving, which would lead to the greatest potential influence for proglacial sedimentation (Brinkerhoff et al., 2017).

The resulting glacier evolution is effectively identical to the experiment without sediment dynamics. This lack of sensitivity is again a result of the basic conclusion that the contemporary lobe extent is incompatible with current climate – the retreat is driven by a surface mass imbalance and as such modification of losses to calving have little influence.

Despite the small modeled influence of sedimentation under this simplified case, Sít' Tlein's current morphology is undoubtedly controlled significantly by the presence of its terminal moraine. Indeed, in the absence of a protective moraine, it is unlikely that Sít' Tlein could have advanced to its current position, and thus it is possible also that Sít' Tlein undergoes tidewater glacier periodicity (Meier and Post, 1987), in which slow sediment-driven advances are punctuated by fast retreats from extended positions. This leads to an alternative or augmenting hypothesis for contemporary surface lowering at Sít' Tlein, namely that the observed surface elevation change signal may be a result not of ice thinning, but rather of fluvial erosion of underlying sedimentary structures. Indeed, observations from nearby Taku Glacier show that contemporary surface lowering rates over the lobe are not inconsistent with observations of the erosion of subglacial sediment (Motyka et al., 2006). Furthermore, Sít' Tlein has exhibited ansynchronous retreat relative to its neighbors, implying that climate forcing alone may not be the sole factor in determining its retreat. On the other hand, observed surface elevation change is relatively consistent across the Sít' Tlein lobe, including over significant subglacial canyons which are unlikely to be of sedimentary origin, which implies that bed elevation changes cannot be responsible for surface lowering there. Exploration of tidewater glacier periodicity in a





large system such as Sít' Tlein and its neighbors via coupling of a realistic model of glacier evolution such as this one to a sediment dynamics model as in (Brinkerhoff et al., 2017) remains an avenue for future work.

## 7.3 Relationship with prior modeling efforts

### 7.3.1 Optimal control

The results in this work combine a variety of numerical and statistical techniques that have been developed in the context of glaciology for over three decades. The adjoint-based PDE-constrained optimization approach for the inference of model parameters from surface observations was pioneered by MacAyeal (1993), who applied such methods to the inference of traction from velocity observations via the shallow shelf approximation. Since that time, such methods have become a workhorse for practical ice sheet modelling, variously being used to determine traction and/or rheological parameters (e.g Joughin et al.,
2004; Morlighem et al., 2013; Sergienko and Hindmarsh, 2013; Habermann et al., 2013; Petra et al., 2014; Isaac et al., 2015; Arthern et al., 2015; Riel et al., 2021, among others). Of particular interest here, such techniques have been used to infer multiple parameters at once; for example, Gudmundsson and Raymond (2008) explored the simultaneous inference of traction and bed geometry from surface expressions using a transfer function approach. Petra et al. (2012) simultaneously inverted for the rheological prefactor and traction coefficient using an inexact Gauss-Newton scheme, while Gudmundsson et al. (2019)
performed a similar inversion across the whole of Antarctica. Ranganathan et al. (2021) simultaneously invert for the traction coefficient and rheological prefactor over ice streams in Antarctica, developing a specialized regularization approach in hopes of finding a unique partitioning between parameters.

### 7.3.2 Time-dependent inversion

While most previous efforts have performed so-called 'snapshot' inversions, in which properties of the diagnostic Stokes'
equations are inferred, the methods described here are closely related to the handful of works that also invert for properties of the mass-continuity equation. Perego et al. (2014) inverted simultaneously for traction and bed geometry so as to match velocities and also produce an ice sheet initial condition that is free from spurious transients; while this inversion was static in time in the sense that it treated the surface rate of change as an observed field that modifies the mass balance, it still incorporated mass-conservation information. Conversely, Goldberg and Heimbach (2013), in perhaps the first application of a
time-dependent adjoint to inverse problems in glaciology, simultaneously inferred traction and bed geometry for a synthetic case from observations of surface velocity and elevation, providing a blueprint for consistent state estimation in ice flow models. Their methods were extended to state-consistent modeling of several glaciers in West Antarctica (Goldberg et al., 2015). Similarly, Larour et al. (2014) combined a spin-up process with the inference of time-dependent surface mass balance and traction fields conditioned on surface elevation and surface velocity observations for the northeast Greenland ice stream, also
employing a time-dependent adjoint. Choi et al. (2023) apply a time-dependent adjoint to a time-series of velocity observations in Greenland.



### 7.3.3 Probabilistic inversion

Most previous works also perform so-called 'deterministic' inversion, in which only the *maximum a posteriori* solution to the associated inverse problem is found. However a growing number of works take a Bayesian approach. The aforementioned Petra et al. (2014) developed a gradient-enhanced MCMC approach that yielded a full posterior distribution. Since such methods do not easily scale to large problems, Isaac et al. (2015) adapted the methods of Bui-Thanh et al. (2013) to the ice sheet context, developing the methodology for the low-rank Laplace approximation to the posterior covariance that we employ in this work. Such methods have been extended a few times since. Koziol et al. (2021) used similar methods to perform a snapshot inversion with estimated posterior covariance of traction for a synthetic case, but also propagated the resulting uncertainty forward through time. Recinos et al. (2023) extended these methods to three West Antarctic Ice Streams. Other works utilize alternative formulations for the approximation of the posterior distribution. Brinkerhoff (2022) used variational inference to determine the multivariate normal distribution that best fits the posterior distribution with respect to the Kullback-Leibler divergence. This work in particular developed the low-rank Gaussian process approach to parameter mdoelling that we adopt here. Riel and Minchew (2023) also employed variational inference in tandem with physics-informed neural networks to infer ice rigidity. Aschwanden and Brinkerhoff (2022) took a different approach, using a neural network to emulate the ice dynamical model, in turn facilitating tractable – if approximate – Markov Chain Monte Carlo sampling of the posterior distribution over a handful of ice flow parameters.

### 7.3.4 Thickness estimation

While we view the geneology of this paper as primarily drawing from the optimal control and Bayesian inference literature described above, one of the principal methodological advances in this work is in the estimation of the ice thickness. BedMachine (Morlighem et al., 2017) used mass conservation *without* Stokes' flow and in conjunction with estimated surface velocities and effective mass balance to infer ice bed elevation; however the resulting product is not necessarily transient-free when combined with an ice flow model due to inconsistencies between model physics and observed velocities. There have also been significant efforts to estimate the thickness of mountain glaciers, often at the global scale. Many such methods – including a distant precursor to the method described here participated in the Ice Thickness Model Comparison Experiments (ITMIX and ITMIX2 – Farinotti et al. (2017) and Farinotti et al. (2021)), which provides an excellent synopsis of such methods.

### 7.3.5 Methodological contribution of this paper

The current paper represents a methodological advance by combining several of the individual lines of inquiry described above. As in previous works, we use an adjoint model for ice flow which allows for the efficient computation of gradients of model inputs with respect to outputs. In contrast, our model is integrated within the general purpose automatic differentiation tool Pytorch, which allows for seamless integration with complex linear algebra operations and differentiation through time. We do note that our method for time-dependent adjoint is not implemented with a strong emphasis on memory savings. For example, we do not implement the checkpoint schemes (Forget et al., 2015; Logan et al., 2020) that are sometimes used in ocean and



ice-sheet adjoint codes. Our choice allows for faster computation, but potentially limits application of our methods to very large systems with long integration times. We have used this tool for the particular purpose of determining a set of optimal model parameters via gradient-based optimization through time (and their associated uncertainty quantification); however, it is quite general and could be used to infer other parameters or establish sensitivities of different quantities of interest with little modification, particularly because the integration with Pytorch makes it straightforward to compute gradients of other non-misfit functions with respect to model inputs (for example, the sensitivity of future grounding line flux with respect to initial conditions).

Also similar to previous efforts, we infer multiple disparate model parameters simultaneously, but in contrast our parameters are time dependent (when appropriate), and represent not only the basal traction and topography, but surface mass balance as well. This simultaneous inference is essential to producing a model that is free of spurious transient dynamics - after all, the configuration of a real glacier represents the long-term integration of ice motion, thickness, and surface mass balance at once, and we have endeavored to reproduce numerically that same physical self-consistency. However, allowing multiple parameters to vary at one also leads to ambiguity in inferred parameters - is a particular observed surface velocity the result of thick ice moving mostly via deformation or thin ice that is sliding faster? As such, we view at least an approximation of posterior co-variance between potentially equifinal parameters as essential. To this end, building on previous works we utilize a well-known method – the Laplace approximation – to derive the uncertainty over inferred parameters. However, here we have extended these methods to operate with a diverse set of spatio-temporal observations and in conjunction with a higher-order time dependent model with both changing velocity and geometry, a first in both cases. More broadly the methodological advances here provide a framework for the creation of ice flow model predictions that accommodate a broad range of observational constraints (and that produce hindcasts that agree with time-dependent observations) while remaining robust to the absence of unknown inputs and quantifying the resulting induced uncertainties.

The inclusion of time-dependent inference and uncertainty quantification is not without cost, and both of these factors lead to significantly increased computational expense relative to a time-static and deterministic inversion. With respect to the former, every evaluation of the likelihood elicits evaluation of the model over the observational period (and perhaps beyond, as in our case). This cannot be reduced because the model must produce a prediction at every time for which data exists. However, it may be possible to significantly accelerate models via emulation, particularly of the stress-balance solver as in Jouvet et al. (2022), although it remains to be seen whether such models are sufficiently robust to handle the diversity of flow conditions in systems such as Sít' Tlein. With respect to uncertainty quantification, both the randomized Laplace approximation and simple Monte Carlo forecasting that we employ here are embarrassingly parallel, a key advantage for computational tractability particularly given the widespread availability of compute cores in the cloud. Nonetheless, the amount of computation needed to accurately characterize inferred parameters our predicted quantities of interest scales with the complexity of the problem, and application of these methods to larger systems like Greenland or Antarctica may require significantly more computation. However, we have been careful – particularly in our construction of parameter representations – to take advantage of approximate sparsity and low-rankness, and we believe that our approach is still reasonable at the ice sheet scale; for reference, the complete computational pipeline for this work took around 20 hours on one laptop computer.





## 7.4 Landscape-scale effects

Our results imply that the continued existence of Sít' Tlein's piedmont lobe is inconsistent with contemporary climate forcing, let alone forcing subject to continued warming. As such, this system will, over the next century, undergo a significant landscape-scale transition from a terrestrially terminating system grounded on a broad array of ice- and non-ice-cored moraines into a lake or ocean-terminating one, presumably with a significant new calving front. The degree to which the resulting system will resemble Bering Glacier to the west, which terminates into a primarily freshwater lake, or Hubbard Glacier to the east, which terminates into a saltwater bay, depends significantly on the potential degradation of the Sít' Tlein forelands.

In either case, the local ecosystem has the potential for significant change. In the near term, enhanced melting and disintegration of the piedmont lobe will lead to an increase in freshwater flux into the Gulf of Alaska. From an oceanographic perspective, such fluxes are a primary driver of coastal circulation, while reduced salinities control along-shore currents resulting from density gradients (Neal et al., 2010). Such modifications to the local biogeochemistry can also have impacts on primary productivity for phytoplankton and the various species that feed on it at various trophic levels, including salmon and marine mammals. From the perspective of physical habitat, the opening of a new coastal bay will also have significant implications on the presence of wildlife. Presumably the disintegration of the forelands will have deleterious effects on local terrestrial ecosystems, including forests growing upon ice-cored moraines and the wildlife populations – such as brown bears – that use them, whereas the increased availability of ice-berg rich waters will provide new habitat for marine mammals such as harbor seals (Blundell et al., 2011). One major impact of piedmont lobe degradation will be the conversion of the terrestrial glacierized landscape–which is part of Wrangell-St. Elias National Park and Preserve, into unprotected marine waters. This could constitute the largest removal of park lands in the history of the National Park System. It is not yet clear what the other potential impacts on human uses, including subsistence use, will be.

## 8 Conclusions

Sít' Tlein, the world's largest piedmont glacier, will with high probability undergo a significant transformation over the next century as its low elevation piedmont lobe disintegrates and transitions into a lake or marine terminating glacier. This conclusion is supported by data-constrained probabilistic ice flow modeling. We used spatio-temporal observations of velocity, radar observations of the glacier bed, a diverse time series of surface elevations, and sparse observations of specific surface mass balance to inform a joint probability distribution over the critical parameters of the SpecEIS ice flow model. Because the system is high-dimensional and the model expensive to evaluate, we apply a number of existing and novel tools to render the probablem tractable: the integration of the time-dependent model with the general purpose automatic differentiation tool Pytorch, careful finite-dimensional representations of model parameters, and low-rank Laplace approximation to the posterior covariance. Our model exhibits very good agreement with observations over the historic record. We then sampled from parameter distributions to drive a model ensemble characterizing Sít' Tlein's future evolution over four centuries. While there is significant spread in total mass loss (between 700 and 1300 gt at 300 years from present), we find the vulnerability of Sít' Tlein's piedmont lobe to be robust to variations in forcing, parameters, and model structure.



*Code and data availability.* Code to run all experiments described in this manuscript, as well as a link to download the necessary data, can be found at https://github.com/CompatibleElementGlacierModel/ManuscriptCode, which will be permanently archived in Zenodo upon acceptance for publication.

**Appendix A: Gradients via the adjoint method**

The adjoint method aims to efficiently compute the gradient of a cost function $\mathcal{L}(\mathsf{V}, \mathsf{m})$ with respect to parameters $\mathsf{m} = [\mathsf{H}_0, \mathsf{B}, \beta, \mathsf{a}]$, where $\mathsf{V} = [H, \bar{\mathsf{U}}, \mathsf{U}']$ is the vector of state variables. We begin by writing a new constrained cost function

$$\mathcal{J}(\mathsf{m}) = \mathcal{L}[\mathsf{V}; \mathsf{m}] + \mathcal{A}[\lambda, \mathsf{V}; \mathsf{m}], \tag{A1}$$

where $\mathcal{A}[\lambda, \mathsf{V}; \mathsf{m}]$ is the constraint and $\lambda = [\lambda_1, \boldsymbol{\lambda}_2]$ are Lagrange multipliers. The constraint here is the semi-discretized weak
form of the coupled forward model (Eqs. 5 and 7)

$$
\begin{aligned}
\mathcal{A}[\lambda, \mathsf{V}; \mathsf{m}] = & \int\limits_{\bar{\Omega}} \lambda_1 \left( \frac{H_{k+1} - H_k}{\Delta t} - \dot{a}_{k+1} \right) \, \mathrm{d}A \\
& + \int\limits_{\partial \bar{\Omega}} [\![ \lambda_1 \, \mathbf{n_x} ]\!] \cdot \widehat{\bar{\mathbf{u}} H} \, \mathrm{d}s, \\
& + \int\limits_{\Gamma_{out}} (\lambda_1 \, \mathbf{n_x} \cdot \bar{\mathbf{u}}_{k+1} H_{k+1}) \, \mathrm{d}s \\
& - \int\limits_{\Omega_T} \nabla_{\mathbf{x},z} \boldsymbol{\lambda}_2 : (2\eta_{k+1} H_{k+1} \dot{\epsilon}_{1,k+1}) \, \mathrm{d}V \\
& - \int\limits_{\Omega_T} \rho_i g \left[ \nabla_{\mathbf{x}} \cdot (\boldsymbol{\lambda}_2 H_{k+1}) \right] S_{k+1} \, \mathrm{d}V \\
& - \int\limits_{\bar{\Omega}_T} \left[ \beta_{t+1}^2 N_{t+1}^p \|\mathbf{u}\|_{k+1}^{m-1} (\boldsymbol{\lambda}_2 \cdot \mathbf{u_{k+1}}) \right]_{\varsigma=1} \, \mathrm{d}A \\
& - \int\limits_{\Gamma_W} \alpha(\mathbf{u} \cdot \mathbf{n_x})(\boldsymbol{\lambda}_2 \cdot \mathbf{n_x}) \, \mathrm{d}s \\
& - \int\limits_{\partial \Omega} \rho_i g [\![ \mathbf{n_x} \cdot \boldsymbol{\lambda}_2 S_{k+1} ]\!] \{ H_{k+1} \} \, \mathrm{d}A,
\end{aligned}
\tag{A2}
$$

where we have used integration by parts to substitute natural boundary conditions, and in which jump, average, and numerical
flux terms appear due to the discontinuous Galerkin discretization of the thickness field – a full description of the above
manipulations can be found in Brinkerhoff (2023)).

We seek to compute the gradient with respect to the parameters by eliminating the state variables and Lagrange multipliers. Taking the first variation (i.e. the Gateaux derivative) of Eq. A1 with respect to the (basis expansion of the) Lagrange multiplier in the direction of a test function and setting the result to zero recovers the weak form of the forward model, which can be
solved via finite elements as usual. Taking the first variation with respect to the (basis expansion of the) model state variables





in the direction of a test function and equating the result with zero yields a weak form of the adjoint equation with right hand side given by $\frac{\partial \mathcal{L}}{\partial V}$, the derivative of the cost with respect to the forward model's output, and precisely the quantity delivered by reverse mode automatic differentiation.

The adjoint equation is structurally similar to the forward equation, with both an adjoint transport equation (which propagates misfit information opposite in time and spatial direction relative to the forward model) and an adjoint stress balance equation (which has a more complex viscosity term). The adjoint equations are linear in $\lambda$ and can be solved with finite elements using similar methods to the forward model.

With forward and adjoint systems solved, the gradient terms can be readily computed by taking the Gateaux derivative of Eq. A1 with respect to $\mathbf{m}$ in the direction of suitable test functions. The resulting expressions will generally depend on both 960 $\lambda$ and $V$ and can be evaluated to find the desired derivatives in terms of the basis coefficients $\frac{\partial \mathcal{L}}{\partial \mathbf{m}}$, which can be used directly or – as in our case – propagated further back in the computational graph to compute gradients of the cost with respect to the arguments of the functions used to form m. We do not calculate either the analytical representation of either the adjoint equation or gradient expressions by hand, instead relying on the symbolic differentiation capabilities of Firedrake.

**Appendix B: Randomized low rank approximation of the posterior covariance**

The posterior covariance matrix emerging from the Laplace approximation

$$\Sigma_{post} = (\mathcal{H} + I)^{-1} \tag{B1}$$

is intractable to compute. To circumvent this, we follow Bui-Thanh et al. (2013) and approximate it with a low-rank eigenvalue decomposition

$$\mathcal{H} \approx V D V^T, \tag{B2}$$

with $V \in \mathcal{R}^{m \times r}$ the eigenvectors and $D \in \mathcal{R}^{r \times r}$ a diagonal matrix containing the leading $r$ eigenvalues of the Hessian.

We use a variant of the randomized methods described in Halko et al. (2011) to form the approximate decomposition. The randomized method proceeds as follows. First, given a low-rank and positive semi-definite matrix, we can write the following approximation

$$\mathcal{H} \approx Q Q^T \mathcal{H} Q Q^T, \tag{B3}$$

where $Q \in \mathcal{R}^{m \times r}$ is an orthonormal basis for the range of $\mathcal{H}$. In an effort to build a randomized subspace for this range, we compute the product $\mathbf{Y} = \mathcal{H} \mathbf{\Omega}$, where $\mathbf{\Omega}$ is a random matrix with entries drawn from the standard normal distribution. Even without being able to directly compute the Hessian, we can compute Hessian vector products using the classic finite difference approximation

$$\mathcal{H} \mathbf{v} \approx \frac{\nabla_\zeta \mathcal{L}(\boldsymbol{\zeta}_{MAP} + \epsilon \mathbf{v}) - \nabla_\zeta \mathcal{L}(\boldsymbol{\zeta}_{MAP})}{\epsilon}, \tag{B4}$$





where $\epsilon$ is a small constant and $\mathbf{v}$ is an arbitrary vector. This HVP is not exact (because finite differences are not exact, nor is our forward solver), which leads to a variation relative to standard randomized methods for computing eigendecompositions. With the sample matrix $Y$ in hand, we can compute the standard QR decomposition

$$\mathbf{Y} = QR \tag{B5}$$

to produce an approximate orthonormal basis for the range of $\mathcal{H}$. We then define the factor

$$B = Q^T \mathcal{H} Q. \tag{B6}$$

left and right-multiplying by $\Omega^T Q$ and $Q^T \Omega$ respectively, we have that

$$\Omega^T Q B Q^T \Omega = \Omega^T Q Q^T \mathcal{H} Q Q^T \Omega. \tag{B7}$$

Using the identity $QQ^T \mathcal{H} QQ^T \approx \mathcal{H}$, we have

$$B = (\Omega^T Q)^\dagger \Omega^T \mathcal{H} \Omega (Q^T \Omega)^\dagger, \tag{B8}$$

which is a square matrix in $\mathbb{R}^{r \times r}$, which can be easily manipulated. This immediately yields the eigendecomposition

$$\mathcal{H} \approx V \Lambda V^T, \tag{B9}$$

where $V = QU$, and $U$ and $\Lambda$ are the eigenvectors and eigenvalues of $B$.

In principle, $B$ should be symmetric and positive definite, but because the matrix-vector products $\mathcal{H}\Omega$ are not exact, this will not necessarily be true. As such, instead of using $B$ directly in Eq. B9, we first symmetrize using

$$B' = \frac{B + B^T}{2}, \tag{B10}$$

and then project B' to the space of positive semi-definite matrices by ignoring its negative eigenvalues and associated eigenvectors. This projection is optimal with respect to the Frobenius norm (Tropp et al., 2017).

With a low rank approximation to the data Hessian, we can form an approximation to the covariance matrix for $\zeta$ as

$$\Sigma_{post} = I - VDV^T, \tag{B11}$$

where $D = \frac{\Lambda}{\Lambda+1}$ and we have used the matrix inversion lemma. For this approximation to be highly accurate, we require that $\Lambda \ll 1$. Nonetheless, even if this condition is *not* met, the resulting covariance will strictly overestimate the posterior variance, since it is formulated as the subtraction of a positive semi-definite matrix - which in some sense represents the data gain - from the prior. This matrix is large, so we never form it directly. Rather, we are interested in two downstream tasks; first, for the purposes of visualizing the posterior uncertainty in the inferred bed, traction, and mass balance, we are interested in the marginal variance for a model parameter at some spatio-temporal point, which can be computed as, e.g.

$$\text{var}[B(\mathbf{x})] = \sum_j \left[ L(\mathbf{x})_j^2 - (L(\mathbf{x})_j V \sqrt{D})_j^2 \right], \tag{B12}$$

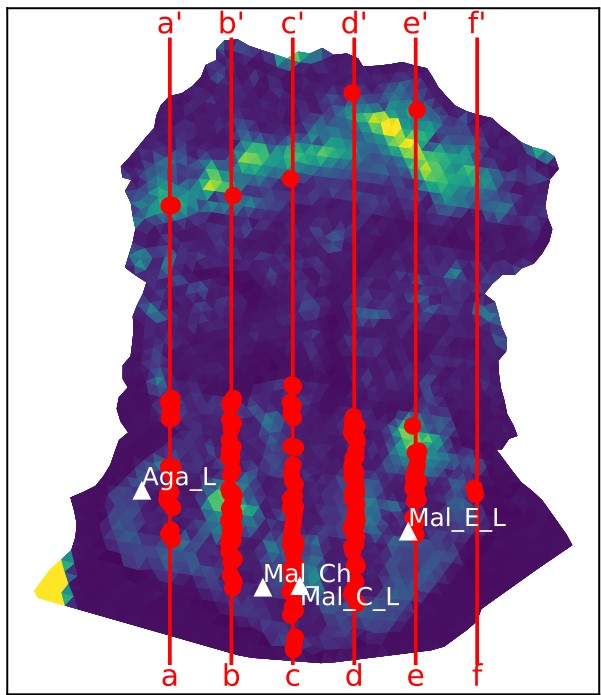

**Figure C1.** Map of posterior standard deviation of the bed elevation when bed data is held out in a 'checkerboard' pattern. Overlain red lines are transects over which we plot the inferred bed using both the full and restricted model. Red points are radar observations that lie within 1km of transects. White labelled triangles indicate the location of ablation measurements over summer of 2021.

where $L(\mathbf{x})$ is the appropriate prior basis computed at $\mathbf{x}$ as described in Sec. 27 and elsewhere. This is computationally tractable to evaluate, as we never need to form the full matrix to evaluate its diagonal. Second, we are also interested in drawing samples from the posterior distribution for the purposes of evaluating downstream sensitivity. Sampling for a multivariate normal

requires a matrix root; fortunately, the particular form of $\Sigma_{post}$ allows for a remarkably convenient computation of a root $GG^T = \Sigma_{post}$ as

$$G = I + V P V^T, \tag{B13}$$

where $P = \frac{1}{\sqrt{\Lambda + 1}} - 1$.

## Appendix C: Comparison of model predictions against unseen observations

In an effort to understand the validity of the bed and surface mass balance fields inferred by our model, we compare them in a few ways to observations.



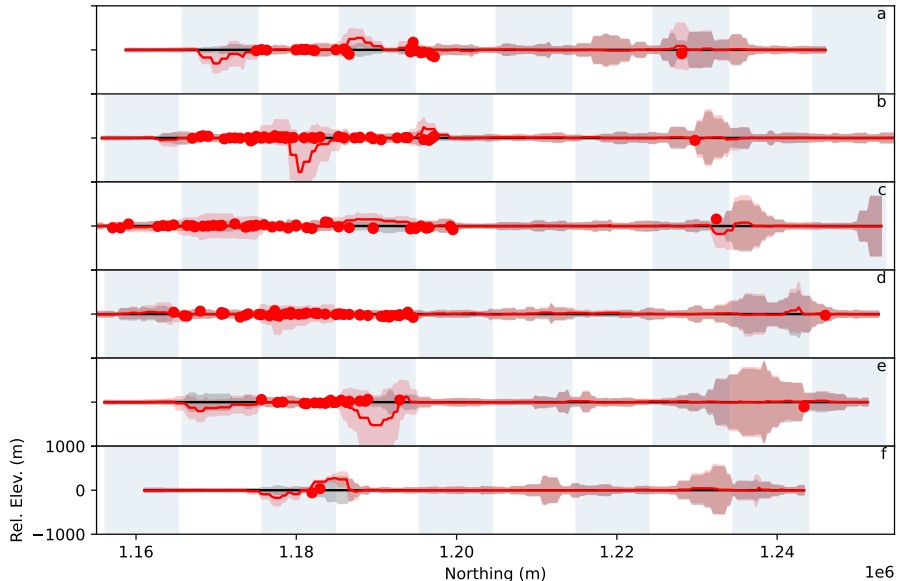

**Figure C2.** Posterior bed estimates along the profiles appearing in Fig. C1. The mean (solid line) and $3\sigma$ credible interval (shading) for both the full (black) and restricted (red) model are shown relative to the full model mean. Red dots indicate radar observations that lie within 1km of the indicated transect. Blue shading implies that the restricted model does not have access to observations lying within.

## C1   Inferred bed versus held-out data

As a first experiment, we assess our model performance in recovering the observed bed elevation when those observations are excluded from the analysis. To do this, we repeat the analysis described in Sec. 4 *de novo*, but eliminate radar-derived bed observations over the whole domain in a checkerboard pattern with a block size of 10km (we keep the bed constraints derived from digital elevation models in unglaciated areas). We then extract the resulting bed elevation prediction and marginal standard deviation over the profiles shown in Fig. C1 and plot these elevations (the restricted model) relative to those inferred when using the complete dataset (the full model, Fig. C2). Additionally, we plot any data point that falls within 1km of the profile. We find that in the accumulation zone, the restricted model does not deviate much from the full model. This is because the IceBridge observations are highly limited in the accumulation area anyways, so the modelled bed is mostly the result of mass conservation anyways. Nonetheless in a few locations, namely in profiles a and c, the restricted model recovers the observed bed even when it is not provided as a constraint.

## C2   Inferred bed versus a new dataset

During the same field campaign in which we collected the surface mass balance cores described in Fig. 2, we also surveyed a small number of profiles using a ground-based radar system, with bed returns manually picked. These observations were not included in the analysis described in the main text – indeed we did not look at them until after the analysis was complete.



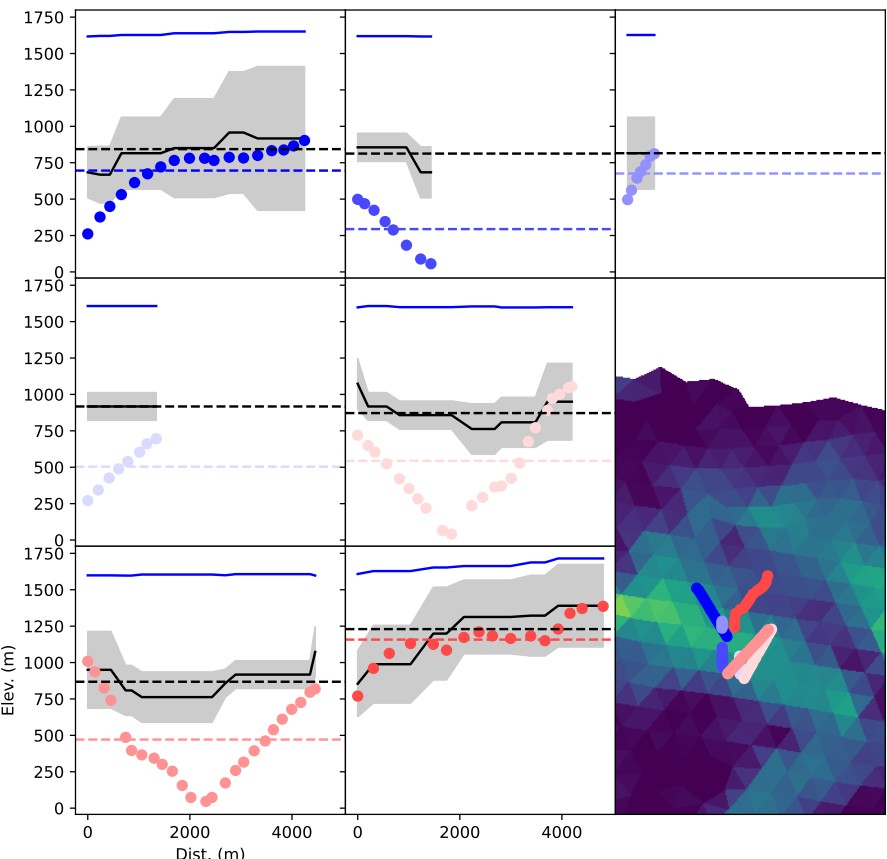

**Figure C3.** Posterior bed estimates at the color-coded transect locations indicated in the lower right plot (above and to the northwest of the Seward throat). Bed picks appear as colored dots. The surface elevation is given by the thin blue line. The modeled mean (solid line) and $3\sigma$ credible interval (shading) are shown in black/gray. Dashed lines indicate modeled and observed means.

The radar-derived bed elevations are plotted alongside (full) model predictions in Fig. C3. We find that in areas that the radar suggests are below 800m deep, the model and observations exhibit surprisingly good agreement, with the observations falling within the model's posterior $3\sigma$ credible interval in most cases. However, the radar returns also suggest the existence of an

exceptionally deep V-shaped trough (on the order of 1600m) that the model does not capture. This disagreement is vexing and points to a physical inconsistency in one of the datasets involved. Because SpecEIS has been verified to conserve mass precisely, the classical glaciological formula of area-integrated mass conservation

$$\int_A \dot{a} - \frac{\partial H}{\partial t} \, \mathrm{d}A = \int_s \bar{\mathbf{u}}H \cdot \mathbf{n} \, \mathrm{d}s \tag{C1}$$

holds, where $s$ is the cross-section and $A$ is the contributing area. Examining this relation, we observe that there are a few

ways in which the model could produce a thickness that is too low. First, the surface mass balance could be underestimated.





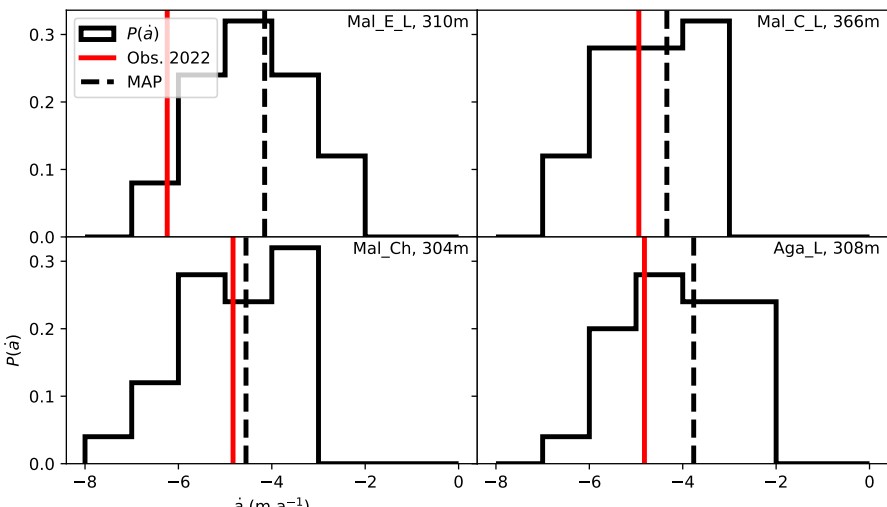

**Figure C4.** Modelled and observed surface mass balance rates for four locations on the Sít' Tlein piedmont lobe indicated in Fig. C1.

While three separate field campaigns spanning 70 years (Sharp (1951), Marcus and Ragle (1970), and the present work) have established approximate accumulation and net balance rates that are consistent with the model, these observations also cannot account for potential internal accumulation, which could be a source of discrepancy – although it is difficult conceptualize a means for meltwater to refreeze in the relatively temperate climate of coastal Alaska. Second, the thinning rate (which the

model successfully reproduces) could be underestimated. Although we expect the laser altimetry from which it is derived here to be accurate and the time differences between measurements are long, there are also potential confounding factors – such as changes in firn compaction rates – that may introduce bias. Third, the ITS_LIVE-based surface velocity observations (which, again, SpecEIS successfully reproduces) could be overestimated, or might not represent the true temporal average which is required here. Indeed, a reproduction of the inference described in the main text with the additional radar-derived

measurements described here *can* reproduce said measurements – at the cost of locally underestimating surface velocities relative to observations. We believe that this is the most likely scenario and hope to perform a more detailed analysis in future work. Finally, it is possible that the radar observations themselves are mistaken – it is notably difficult to obtain high-quality bed returns in extreme topography due both to clutter from off-nadir returns and the attenuation of signals in thick, temperate ice, which is why the IceBridge dataset contains no observations for this location to begin with.

While the disagreement described above remains a mystery, it seems to be relatively localized and we do not expect this misfit to materially alter the conclusions of this paper, particularly since the area undergoing the greatest change – the piedmont lobe – is extremely well-constrained.



## C3 Inferred ablation rates versus partial observations

During the summer of 2021, we collected melt measurements between June 4th and August 31st at four locations at approximately 300m on the Sít' Tlein lobe (shown in Fig. C1), capturing the majority of the melt season – although these measurements are likely to be an underestimate of the true melt. The posterior distribution of model predictions are shown alongside these observations in Fig. C4. While the model's predicted mass balance is quite uncertain (because we allow for annual noise in the prediction), the magnitude of the melt is in reasonable agreement with observations. It is worth noting that we did not explicitly impose *any* constraints on the surface mass balance below the ELA in this analysis – the recovery of rates that are reasonable with respect to observations occurs solely because such rates are required to reproduce the observed geometry and its rate of change.

*Author contributions.* DB, JH, CL, MF, KT, ML, and MT conceptualized and acquired financial support. BT, MC, JH, CL, and MT collected and processed surface and bed altimetry. MD and JH collected and analyzed airborne accumulation radar. VD and MT collected ablation measurements. VD and MF developed and analyzed surface velocity observations. DB, MD, VD, and MT collected snow cores and ground-based radar. DB developed the ice flow model. DB and BT developed numerical methods. DB wrote the original draft, and all authors contributed to revisions.

*Competing interests.* The authors declare no competing interests

*Acknowledgements.* DB was supported by NSF Grant No. 1929718. BT, MD, and JH were supported by NSF Grant No. 1929577. VD, CL, MF, KT, RM, and MT were supported by NSF No. 1929566. We thank Anna Thompson, Sydney Mooneyham, Tyler Kuehn, Natalie Wagner, and Annegret Pohle for helpful discussions. We thank Icefield Discovery for field logistics. We thank the editor Cheng Gong for comments that improved the quality of the manuscript.



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
