# Peer review of "The demise of the world's largest piedmont glacier: a probabilistic forecast"

_EGUsphere, 2024_

## Referee Comment (RC1)

**Review of 'The demise of the world's largest piedmont glacier: a probabilistic forecast' by Brinkerhoff et al.**

**General Comments**

This manuscript is interesting, novel, and well written. It includes interesting experiments related to the probable future development of Sít' Tlein which are of interest to the cryospheric community.

However, I have several comments which I believe could help improve the manuscript:

1. *How does the methodology used here impact results?*

The method used in this manuscript is novel and certainly interesting/useful, but in L880 you state 'The inclusion of time-dependent inference and uncertainty quantification is not without cost, and both of these factors lead to significantly increased computational expense relative to a time-static and deterministic inversion'. It is hard to ascertain from the manuscript whether the additional computational cost leads to a significant difference in results. It would be great if there was more explicit discussion of why and when your methodology is the best choice, or if the same future scenarios could be run from a 'traditional' initial state to allow for a comparison to be made. Without this, it is hard to fully assess the use of the methodology.

2. *Some moving around of material may be beneficial for readability*

The manuscript is long, which I realise is somewhat necessary to fully explain the methodology. However, it sometimes felt like I was swapping between two manuscripts; one on model development, and one on the future of Sít' Tlein. It could be beneficial to move some of the material around so that the manuscript first introduces the new methodology, how it is implemented, and how it constitutes a methodological advance. Then, the case of Sít' Tlein could be presented to demonstrate the use of the methodology (ideally with a comparison to experiments described in point 1). I think this is already the approach you have gone for, but could be more robustly applied.

**Minor and technical comments**

Fig.1 – Panel c appears to be missing but is referenced in the caption

L38 – Why 2344?

L108: Can you please describe your motivation for choosing a Budd law

L126: So only one vertical layer in the model?

L140: What other mesh resolutions did you try

Fig. 3 – Same colour scale for b and d? Although I realise this may make the data too difficult to see. . .

L590: Linear increase in mean air temperatures – what is the justification for this linear increase? How does this compare to e.g. different SSPs for the future?

Fig.4 : The caption refers to panels a,b and c – but this notation is not shown in the figure.

L684: Maybe it comes later, but it is interesting that there is a limited qualitative

difference between scenarios in 2073 – it would be interesting to discuss this further (e.g. how much warmer/more negative is the projected vs fixed SMB at this time point?)

Sect. 7.1.2 and 7.1.3: Have any SMB fields generated via other methods (e.g. RACMO/MAR, PDD model) been created for this region? It would be interesting to have a visual comparison

L793: I think this is a really interesting point (that you have found SMB to be the driver of retreat, but that climate forcing alone may not explain its retreat as it has behaved asynchronously to neighbouring glaciers) - maybe you can expand a bit more on this

Sect. 7.3.1 to 7.3.4 – These subsections contain very little discussion, and are more of a description of previous modelling efforts. Whilst I see that the discussion comes in sect 7.3.5, it would be beneficial to weave in discussion points alongside the description of past efforts (or move these sections to e.g. after the introduction as a background on the methods applied/developed by you?)

L874: Should this be in methods?

L880: To assess whether this computational expense is 'worth it', we ideally need to see an example of the same forward experiments run with a traditional approach

L895: It feels odd that this section comes after the discussion on model developments, when other discussion sections relating more specifically to Sít' Tlein come earlier

L910-913: Interesting!

---

## Author Comment (AC1)

**Author response to reviewer comments**

Doug Brinkerhoff and co-authors

December 21, 2024

We are grateful for the exceptionally helpful comments provided by both the Scientific Editor Cheng Gong, as well as two anonymous reviewers. It has been a pleasure to respond, and we hope that these modifications have lead to a manuscript that is easier to understand, both with respect to its scientific implications as well as the methodological approach and its implications.

Reviewer comments below are in blue, while our responses are in black.

**Response to Reviewer 1**

**Main comments**

How does the methodology used here impact results? The method used in this manuscript is novel and certainly interesting/useful, but in L880 you state 'The inclusion of time-dependent inference and uncertainty quantification is not without cost, and both of these factors lead to significantly increased computational expense relative to a time-static and deterministic inversion'. It is hard to ascertain from the manuscript whether the additional computational cost leads to a significant difference in results. It would be great if there was more explicit discussion of why and when your methodology is the best choice, or if the same future scenarios could be run from a 'traditional' initial state to allow for a comparison to be made. Without this, it is hard to fully assess the use of the methodology.

Thank you for this comment - unfortunately, credibly modelling this system without the methodological advances presented here is not possible. This is partially (although not entirely) a function of the severe data limitations in this system: we do not know the bed elevation (unlike, say, Greenland, where this has been strongly constrained by flightlines), the surface mass balance (which again, in e.g. Greenland is reasonably well-constrained by reanalyzed regional climate modeling, even as a function of time), nor the basal traction (which is typical everywhere). In particular, the requirement that we infer the time-varying SMB necessitates the use of time-dependent inference. Observed changes in surface elevation provide the primary constraint on the way in which SMB varies through time, yet this signal is in a relative sense very small and is easily overwhelmed by so-called 'initialization shock' induced by the misalignment between (imperfect) model physics and (imperfect) observations.

We admit to being a bit stuck with respect to how to modify the manuscript in response to this point, given that we don't have the capacity (or the ability) to develop a different method to act as a straw man. In rewriting the discussion, we have done our best to emphasize how the present work relates to 'traditional' methods, and we hope that this makes it more clear as to why we made the methodological choices that we did.

Some moving around of material may be beneficial for readability. The manuscript is long, which I realise is somewhat necessary to fully explain the methodology. However, it sometimes felt like I was swapping between two manuscripts; one on model development, and one on the future of Sit' Tlein. It could be beneficial to move some of the material around so that the manuscript first introduces the new methodology,

how it is implemented, and how it constitutes a methodological advance. Then, the case of Sit' Tlein could be presented to demonstrate the use of the methodology (ideally with a comparison to experiments described in point 1). — think this is already the approach you have gone for, but could be more robustly applied.

Thank you for this comment, which is consistent with feedback from the other reviewer as well. We have taken a slightly different approach than what is suggested here, but one that we think at least partially remedies the interleaving of scientific and methodological sections. Specifically, we have opted to move technical sections to referenced appendices, while maintaining a plain(er)-language summary in the main body of the text. This has of course made the main manuscript shorter and much more oriented towards the scientific conclusions, while still maintaining detailed methodology.

**Minor and technical comments**

Fig.1 — Panel c appears to be missing but is referenced in the caption Panel c is the small inset of Alaska in the upper corner. We have changed the inset color to make this more visible, and stated what and where it is in the caption.

L38 — Why 2344? The reason is a little bit arbitrary - we have 36 years for which we infer the basal traction, and so multiplied that by 10 to get 360 years. Beginning from 1985 (and using zero based indexing), we get 2344. However, we don't think that this is any more arbitrary than picking round base-10 numbers like 300 years, etc.

L108: Can you please describe your motivation for choosing a Budd law Yes. We have added the following to the manuscript: This model of water pressure is motivated by the observation that water flows out of the terminus of Sít' Tlein at sea level, which places a lower limit on the water pressure beneath the glacier, alongside water pressure measurements from boreholes across many temperate glaciers suggesting a high fraction of overburden as typical. Deviations from this mean-field approximation are subsumed into the specification of the traction coefficient $\beta$.

L126: So only one vertical layer in the model? Yes, but that one vertical layer is more expressive than a multilayer model discretized with a simpler ansatz. The referenced paper demonstrates a better-than-favorable comparison between this discretization choice and a more classical Blatter-Pattyn discretization.

L140: What other mesh resolutions did you try 750m. We have clarified this in the text (in Sec. 7.2.1) as: Finally, our spatial discretization of the model physics is fairly coarse, which was necessary for computational efficiency. We performed some qualitative experiments to assess the impact of this. In particular, we performed the time-independent phase of the MAP estimation using a mesh with nominal 750m resolution. We also ran one member from each of the ensemble experiments above with this high-resolution mesh (using parameters inferred from the lower-resolution version). In each case, we did not find significant qualitative differences. However, this does not constitute a full-convergence analysis, and the influence of higher-resolution meshes remains a topic of future study.

Fig. 3: Same colour scale for b and d? Although I realise this may make the data too difficult to see... We would have preferred this, and it was our initial approach. However, the mismatch between the ranges of the prior and posterior made it so that one or the other would saturate. We will leave the figure as is, but include text in the caption ensuring that readers notice the difference in scale.

L590: Linear increase in mean air temperatures — what is the justification for this linear increase? How does this compare to e.g. different SSPs for the future? Assuming that the time-variation in our inferred surface mass balance rates principally aliases temperature, then our upper-end projections (which assume that the future rate of change in surface mass balance mirrors that of the present) correspond to a climate scenario in which future rates of warming also largely mirror contemporary rates. With respect to RCP scenarios, this most closely corresponds to RCP6.0, and so represents a high - but feasible - melt scenario. See also our response to Reviewer 2, who had a similar comment. We have included explanation of this in the text.

Fig. 4: The caption refers to panels a, b, and c — but this notation is not shown in the figure. Since the figures are already referenced and labeled by their dates, we've removed the reference to sub-panel letter labels.

L684: Maybe it comes later, but it is interesting that there is a limited qualitative difference between scenarios in 2073 — it would be interesting to discuss this further (e.g., how much warmer/more negative is the projected vs fixed SMB at this time point?). We agree. We have added language in this section describing the absolute and relative differences in surface mass balance at elevation and sea level, and described how these differences, when integrated over 50 years leads to little relative change.

Sect. 7.1.2 and 7.1.3: Have any SMB fields generated via other methods (e.g., RACMO/MAR, PDD model) been created for this region? It would be interesting to have a visual comparison. Unfortunately not to our knowledge, hence our relatively unsophisticated approach to the dynamical modelling of SMB - this really is terra incognita in a lot of ways.

L793: I think this is a really interesting point (that you have found SMB to be the driver of retreat, but that climate forcing alone may not explain its retreat as it has behaved asynchronously to neighbouring glaciers) — maybe you can expand a bit more on this. Asynchronicity seems to be common in Alaska coastal glaciers due to the tidewater glacier cycle, and we believe that that is the principle driver of the St. Elias glaciers' broad configurations. We have modified this discussion in the paper to the following: Furthermore, Sít' Tlein has exhibited ansynchronous retreat relative to its neighbors such as Hubbard Glacier, which retreated and began re-advancing in the early 19th century, and the collective terminus of Tsaa, Guyot, Yahtse, and Tyndall glaciers, which filled contemporary Icy Bay and began a retreat towards its present configuration around the turn of the 20th century. Such asynchronicity implies that climate forcing alone may not be the sole factor in determining its retreat. On the other hand, observed surface elevation change is relatively consistent across the Sít' Tlein lobe, including over significant subglacial canyons which are unlikely to be of sedimentary origin, which implies that bed elevation changes cannot be responsible for surface lowering there. Detailed exploration of tidewater glacier periodicity in a large system such as Sít' Tlein and its neighbors via coupling of a realistic model of glacier evolution such as this one to a sediment dynamics model as in [citation] remains an avenue for future work – however, we think it likely that the contemporary configuration of St. Elias Range glaciers broadly reflects long time-scale sedimentary processes driving natural variability, on top of which is superimposed a strong contemporary warming signal.

Sect. 7.3.1 to 7.3.4: These subsections contain very little discussion, and are more of a description of previous modelling efforts. Whilst I see that the discussion comes in Sect. 7.3.5, it would be beneficial to weave in discussion points alongside the description of past efforts (or move these sections to, e.g., after the introduction as a background on the methods applied/developed by you?). We have taken the first suggestion, and modified this part of the discussion to interleave both the previous works as well as their relationship to the methods described in this paper instead of presenting then separately.

L874: Should this be in methods? We are not sure we entirely understand the comment - we aren't including any new methods here, just referring to Sec. 4.6 here.

L880: To assess whether this computational expense is 'worth it,' we ideally need to see an example of the same forward experiments run with a traditional approach. Our goal in this paper is not to produce a methodological inter-comparison (which is another reason that we are reticent to split the paper into two manuscripts), and we would be unable to meaningfully do so. Indeed, it would be very difficult to determine specifically what a 'traditional' approach entails – however, one such option might be as follows: first, assume an arbitrarily determined bed elevation (perhaps taken from a global estimation study using mass conservation that we already know to be erroneous for this system, see Tober, 2023). Second, assume a thickness induced by the 2013 Copernicus DEM, because this is the only product that covers the complete glacier domain. Third, under the previous assumptions invert for basal traction. Finally, force the model into the future using precisely one climate realization taken from a bias-corrected reanalysis model that does resolve the topography and for which the nearest weather station is 60km away and at sea level. While such

a work flow is not uncommon, we would be hard pressed to argue that it is defensible in this case where all such inputs are unknown and we would expect the errors to compound. Even if the results of such an exercise were consistent with those presented here we could not assert that it was so because of any physically robust principles. This is why developed the methodology that we present in this paper: we view it not as valuable because of its relative complexity compared to a methodology such as the one described above, but because it is the simplest work flow that we could devise that credibly addresses the fact that 1) we don't know the bed, 2) we don't know the climate forcing, 3) we don't know the basal properties, and 4) the physics are imperfect, but yet we need to make predictions. We think it is self-evident that these predictions ought to at least agree with the limited set of observations that we do have. Of course, there may be more than one way to obtain this agreement, and these different plausible configurations might have salient differences when extrapolated into the future – this is why it is essential to proceed probabilistically.

L895: It feels odd that this section comes after the discussion on model developments when other discussion sections relating more specifically to Sit' Tlein come earlier. Agreed. We have moved this section to the beginning of the discussion rather than the end.

L910-913: Interesting!

**Response to Reviewer 2**

**Main comments**

This is a review of the manuscript by Brinkerhoff et al. on "The demise of the world's largest piedmont glacier: a probabilistic forecast", submitted to The Cryosphere. This study is focused on Sit' Tlein, the world's largest piedmont glacier. It describes the methodological advances needed to estimate time-dependent glacier model parameters for Sit' Tlein over the historical period, and then to produce a probabilistic forecast of it's future evolution. During the historical period, a range of observations have been made which are used as the inputs to a Bayesian method which efficiently estimates the posterior distribution of model parameters which are then used in future forecast simulations.

This is an impressive study, and one which clearly brings together many years of work on observational and modeling techniques to produce efficient methods for Bayesian parameter estimation and probabalistic forecasting. This style of modeling is at the forefront of ice sheet modeling, and as such this study represents a valuable and important benchmark for the ice sheet modeling community, and how we "should" be doing things.

I have a number of technical questions about the methods that came up as I read through the manuscript. These are detailed below. First, I detail some of my broader concerns with the manuscript and its structure, and a few bigger choices.

1. Perhaps my biggest comment is that the shear length and density of this manuscript may be a barrier to more widespread engagement with this important work. The paper almost read as two papers to me: a methods paper and a modeling paper. While many of the individual methodological steps have been detailed in prior publications, sections 2 and 3 and parts of 4 (constituting about half the manuscript) put all these pieces together to describe what is effectively (for lack of a better descriptor) a data assimilation workflow for glacier modeling. While I think these details are very interesting, that may not be true of the entire audience for this paper, particularly with the eye-catching results that come out of this method. I think there are two options to address this issue: (1) the easiest thing to do would be to move the bulk of the methodological detail into the appendix and instead just provide a high-level overview in a single short section in the main text, with heavy reference to the appendix, or (2) there could be a whole second paper just on the methods, appropriate for a journal like GMD or JAMES.

We very much appreciate this comment, which is consistent with those of the other referee and the SE. We have thought about this issue quite a lot, and ultimately we feel that it is important to keep the paper as

one manuscript. The reason for this is that our methodological choices (though applicable to other places, of course) are principally motivated by the challenges that we have encountered in trying to make projections for this system (which sometimes feels diabolically pathological, due to the convergence of extreme topography, extreme climate, oceanic interactions, severe data limitations, and size). We would be hard-pressed to derive - say - a synthetic example that could illustrate the benefits of the composition of the methods described here as well as does the case for which they were developed. Simultaneously, we are acutely aware that despite our best efforts, the work presented here is imperfect. We want to make sure that the reader has the opportunity (and is perhaps encouraged) to consider our assumptions while contextualizing our results, and we view the inclusion of the technical details in the manuscript (and not in a reference to a companion paper) as contributing to that end. Nonetheless, we agree with the reviewer that a reorganization could allow easier engagement with the manuscript, particularly for those who are more interested in the science implications than the methodological ones. With this in mind, we've take the reviewer's first suggestion, and developed a comprehensive set of appendices that comprise the bulk of the methodological details, which are referenced from plain(er)-language summaries that appear in the main body. While the manuscript is still long (longer now, in fact), we hope that this change adequately addresses the accessibility concerns of all referees.

2. This study was incredibly meticulous in constructing a state-of-the-art method for accurate Bayesian parameter (and state) estimation for the hindcast period. It was then surprising that a few highly simplistic choices were made in configuring the forecast simulations that were only lightly justified or explored. There are two in particular that it would be useful to understand in more detail:

(a) SMB forcing - the "control" case is a reasonable end member, but I'm not sure what to make of the linear extrapolation. You show that while most of the piedmont lobe is effectively "committed" to future mass loss based on current climate, there is a large difference between the present and projected climate states (comparable or larger than posterior parametric uncertainty projected from hindcast). It thus bears more justification that this continued rate of SMB change is comparable to what is actually projected for this region. Polar temperatures are warming faster than the global average, and in some places accelerations are expected in the future. It would be nice to have maybe a high end-member based on some reasonable realistic projection of SMB changes in this region due to a high-end emissions scenario. As you indicates, its easy enough computationally to run a new set of simulations sampling the posterior, so it shouldn't be too challenging to do this.

This is a reasonable concern, and also one that the first reviewer had. It is, of course, difficult to unambiguously draw a 1:1 connection between projected future warming (which is derived from ensembles of highly complex and still fairly uncertain climate models) and our (or any) parameterization of surface mass balance (which is data-driven). However, if we make the simplifying assumption that the relationship between $\Delta$ SMB and (an appropriate spatiotemporal mean of) $\Delta T$ is linear, then of course our assumption of an SMB that changes in the future at the same rate that it has in the recent past would be consistent with a scenario in which temperature also increases linearly at the same rate as the recent past. This slope-matching does in some sense 'bake in' the fact that perhaps local temperature changes deviates from global means. Analyzing, for example, the CMIP6 ensemble, such linear extrapolation of temperature roughly corresponds to RCP6.0 at 2100, and slightly exceeds the median prediction of RCP8.5 at 2300. Taken together, because our data-driven approach already includes historic temperature amplification effects in the calibrated SMB, and because the linear extrapolation approach (when taken alongside some simplifying assumptions about the relationship between temperature and SMB) corresponds to upper plausible end member scenarios, we think that it does represent a reasonable high-end scenario. Testing more specific assumptions would require a means to map from temperature projections to surface mass balance, something that is challenging in the best case and with state of the art tools (see for example this recent paper for Greenland) – neither of which is the case here. We have included language to this effect in the section of the manuscript describing this experimental design (along with a fuller description of its potential shortcomings).

(b) Basal traction - I like the attempt at representing the surge cycle by repeating the time-dependent basal

traction estimate. However, I have no sense if this is actually important. I don't really see any evidence for periodicity in the forecast, so I wonder how much the basal traction variability really matters for the forecast. This is easy enough to test: simply do further forecasts where the basal traction is held constant to values at 2023, and perhaps another for values held constant to something around the late 1980's, since these clearly capture periods with very different velocities along the centerline. In short, it doesn't really matter, particularly when comparing simulations generated using the mean traction with those using the full surge cycle – although this isn't something we really knew until completing the analysis. We have overlain in Figs. 11 and 12 the extent of a projection performed with the average traction. We have also included a section in the discussion that interprets this results (similarly to the section on sedimentation).

3. I say this down below as well, but something I was wondering about throughout the first half of the manuscript is whether this method strictly constitutes a parameter estimation problem or whether direct state estimation occurs as well? It would be helpful to be explicit about exactly what things are being estimated and which aren't. Perhaps a list/table. We do not perform direct state estimation in the sense of a reanalysis product. In fact, we are opposed to directly specifying state (really just the ice thickness in this context) via a mechanism other than generating it with the model itself opens the door to non-physical transient signals – something we try very hard to avoid here. This is especially problematic for glaciers because the response time-scales are so long relative to the period over which data exists. We instead control parameters, which leads to state that agrees with observations to the extent that the physics allow it to. In this sense, our method is similar to the ocean modeling product ECCO. We have added language to this effect to the manuscript: We do not use these observations to directly modify the evolving ice sheet geometry (as might be done in a reanalysis), so all such predictions are derived from a free-running model in a physically self-consistent way, similar in spirit to the oceanic inference engine ECCO [citation]. Regarding a list/table, we argue that Eq. 1 already serves this purpose - however, we hope that the inclusion of the text above makes this clearer.

**Minor and technical comments**

L12: Can you make this statement of "between 500 and 1000" more precise? What are the probabilistic bounds and what are the exact numbers? Perhaps just give the 95% credible range to be more specific. We sure can. We've changed the text to read: If warming ceases and surface mass balance remains at 2023 levels, then by 2073 (2173) we forecast a mass loss (expressed in terms of 95% credible interval) of 323–444 $km^3$ (546–728 $km^3$). If instead surface mass balance continues to change at the same rate as inferred over the historical period, then we forecast a 2073 (2173) mass loss of 383–505 $km^3$ (740–900) $km^3$.

L22: The total area Fixed.

L25: is thinning Fixed.

L27: removed from the coast We think it's okay how it is.

L30: one point connected to piedmont We prefer the present wording, since it is specific to geography.

L53: It would be useful to be clear about whether this is strictly a parameter estimation problem, or a joint parameter and state estimation problem. It was a little unclear to me at this point in the paper. Please see the response to the main point above.

L94: equations; Fixed.

L110: Can you give a bit more justification of this 80% or SL water pressure configuration for effective pressure? This model of water pressure is motivated by the observation that water flows out of the terminus of Sít' Tlein at sea level, which places a lower limit on the water pressure beneath the glacier, alongside water pressure measurements from boreholes across many temperate glaciers suggesting a high fraction of overburden as typical. Deviations from this mean-field approximation are subsumed into the specification of the traction coefficient $\beta$.

L140: Perhaps add these convergence experiments to the supplement. We did not perform formal convergence experiments so to speak, as we would have preferred to use a higher-resolution all things being equal. Rather, we performed a limited subset of qualitative experiments (particularly by running forward simulations using the inferred parameters from lower resolution simulations) and found little difference. We prefer to avoid delving into a detailed examination of the influence of discretization choice, and so we have removed this line, and instead added the following to the discussion of potential shortcomings in Sec. 7.1.1: Finally, our spatial discretization of the model physics is fairly coarse, which was necessary for computational efficiency. We performed some qualitative experiments to assess the impact of this. In particular, we performed the time-independent phase of the MAP estimation using a mesh with nominal 750m resolution. We also ran one member from each of the ensemble experiments above with this high-resolution mesh (using parameters inferred from the lower-resolution version). In each case, we did not find significant qualitative differences. However, this does not constitute a full-convergence analysis, and the influence of higher-resolution meshes remains a topic of future study.

L199: You talk about this simplification a bit more later, but is there a technical/computational reason why you can't treat bed elevation the same as basal traction (i.e. in a time-dependent fashion) in the parameter estimation problem? Wouldn't this address the concern you bring up later about tidewater sedimentation? There is in principle no reason why this isn't possible, but we choose here to make the simplifying assumption that the bed doesn't change because in practice modelling a spatio-temporal field induces a very large number of parameters. We also think that this would add a lot of uncertainty to the inference, since the surface elevation changes could be directly accommodated either by thinning or bed lowering, which would essentially preclude trying to infer surface mass balance.

L200: (Not here, but I thought of it here) Would be useful to cite some of MacKie's papers on geostatistical emulation of bed topography. We are not sure that such a reference would be constructive, since (a) people have been using kriging to estimate topography for a long time, and (b) the use case there is quite different, with the principle utility being the influence of small scale variability on water routing.

L212: This implicitly makes an assumption about smoothness then? Perhaps state explicitly what that assumption is. The smoothness assumption is already explicitly stated on L194. This section describes the construction of a more parsimonious basis for functions which exhibit said smoothness.

L236: Does this mean that only covariance with neighboring elements is preserved? Would this have grid-dependence issues? It does not. The mathematics of Structured Kernel Interpolation is kind of funny – what is being interpolated here is not the function itself, but rather its covariance matrix. Let $k(x, x')$ be a kernel function. If we sample this function at some set of points $\mathbf{x}_1$, then we get one row of the covariance matrix $k(x, \mathbf{x}_1)$. If want to approximate this row at a different set of points $\mathbf{x}_2$, one approach would be to directly compute $k(x, \mathbf{x}_2)$, but an alternative approach would be to approximate it as

$$k(x, \mathbf{x}_2) \approx k(x, \mathbf{x}_1)W^T,$$

where $W^T$ is an appropriately constructed interpolation matrix. In our case, this is good to do because we can compute $k(x, \mathbf{x}_1)$ cheaply and easily due to its assumed structure (namely that the points $\mathbf{x}_1$ live on a regular grid). As long as the interpolation is good (which it is by construction), then the approximate covariance is very close to the true one.

L258: In Tober Fixed.

L345: Be more specific about which geometric observations. This was poorly worded – in reality we try to infer it from *all* observations, which don't necessarily have to just be geometric - velocities, thickness through time, and direct measurements all constrain this function. We have changed this line to read: attempt to infer it from observations in tandem with the model's other parameters

L420: This spin-up procedure makes me curious why you didn't use historical observations directly in the estimation procedure? The error would be large, but you are implicitly using them here, and you already have the framework to incorporate them directly. This is a reasonable idea. We did not do this because we

did not have access to these maps in a digitized or geo-referenced form. However, there is not specific reason why this could not be done and it could potentially be very interesting to do so.

**L425: Is it possible hysteresis could be going on here? I think there needs to be a stronger argument here.** We have justified this more fully with the following text: This initial condition does not, in principle, influence the final steady-state solution that we use as the initial condition for further simulations, so long as the steady state solution is unique. This is the case for terrestrial glaciers (assuming a constant traction), where the mass balance uniquely specifies the ice extent. This may not be true for marine glaciers - for instance initializing Sı't' Tlein from an ice-free state could preclude advance across the submarine basin to the present terminus position because of calving or flotation. However, because the 1915 and 2013 glaciers are both in extended configurations, our optimization procedure only explores the 'extended' branch of this bifurcaction, which effectively behaves as land-terminating.

**L440: Offers complete coverage.** Changed.

**L448: Does interpolation induce correlation?** Yes, by definition, but the character of that correlation is dependent upon the choice of interpolant and the length scale over which it is evaluated.

**L490: Explain why the log posterior is more convenient.** It is more convenient to work with the logarithm of the posterior distribution, both for numerical reasons (e.g. because it is less likely to over- or underflow) and symbolic ones (i.e. the chain rule of differentiation is easier to apply to a sum than a product). Because this function is monotonic, it induces no loss of information.

**L517: Unconstrained or underconstrained?** Unconstrained in the sense of bounds on the variables, an explanation of which we have added parenthetically.

**L565: I think the equation reference here is incorrect.** Indeed, it should reference Eq. 3.

**L632: That are associated.** Fixed.

**Figs 6-7: These should probably have different colormaps, and the Fig 6 colormap would be more intuitive if thinning was red and thickening was blue.** We're not sure that we agree on either point (divergent colormaps are tricky!), but we have included these modifications.

**Fig 9: It would be more intuitive if the points were colored by their corresponding cross-section.** Unfortunately, it doesn't work to have the lines and points for a particular cross-section the same color. We've tried a variant with the data colored and the model as black lines, which we think does look better.

**L658: Where faster flow** Fixed.

**Figs 10-12: I think it would make more sense to plot these as change in thickness. For Fig 10, from 1915, and for Figs 11-12, from 2023.** In principle, we really like this idea, but when we tried it the visualization was a bit underwhelming: the map either reflects the smooth surface mass balance signal in regions of the domain that remain glacierized, or show the (negative of) the thickness in regions where the ice has disappeared. The effect is jarring and feels less instructive than showing the geometry directly, which to our sensibilities, provides a bit more intuition into the glacier's projected configuration.

**Fig 13: I think the change in x-axis scale is a bit more confusing than illuminating, and I was looking for an explanation of the apparent acceleration until I read all the way through the caption. I would instead make the x-axis have a single time scale and then plot an inset in the lower left corner, which is empty anyway, with the zoomed-in period up to 2060.** We agree regarding the axis change. We've switch to the simpler plot. We did not do the inset, because it ends up showing about the same amount of detail as the main plot itself because of its smaller size!

**L697: Another way to say this that might be more straightforward is that changes in the piedmont ablation zone are largely already committed by climate change pre-2023 and changes in the accumulation zone are still largely dependent on future climate change.** We like this description, and have included the following paraphrase as a summary to this paragraph: In summary, mass loss over the piedmont is largely already

committed due to warming that occurred prior to 2023, whereas potential changes in the accumulation zone are still largely dependent on the degree to which climate changes in the future.

L703: I'm not sure if you can say this is "significantly" greater as they are within +/-2 sigma. Of course. We have tried to excise this word from our writing when used informally, but it creeps in sometimes. We have deleted it.

7.1.3 and 7.1.4 section titles: Should say "surface mass balance." Fixed.

L732/736: Need more citations of these points presented as facts with little support. We are surprised that these statements might be viewed as controversial. We have changed the language to try to more clearly express our meaning, and have added a citation regarding degree-day models: This captures some phenomenological features that are observed and that also sometimes show up in other models. For instance, such a model can represent the different relationship between SMB below and above the ELA induced by differing physical processes of melting ice versus snow ?. It also parameterizes the typical rarefaction of precipitation at very high elevations.

L762: It would help to cite Fig 13 here which shows this exact point. Done.

L777: Isn't it the case that this won't really affect the glacier when it is monotonically retreating? Since this modification to bed elevation only occurs in places not in contact with the ice base? Or is there something subtle about floating ice that I'm missing here? In any case, more explanation of how this affects the glacier in this particular case would be useful. Floating or near-floating ice can push against the moraine, and the longitudinal stresses so induced can forestall retreat (at least this occurred in (?). Stated alternatively, this forces the creation of a permanent pinning point that moves along with the glacier. We've added the following lines: We apply this experiment to the present-climate case without calving, which would lead to the greatest potential influence for pro-glacial sedimentation; the sediment can affect ice dynamics by creating compressive longitudinal stresses as newly floating ice pushes against recently deposited 'sediment' as the ice begins to retreat, a mechanism that we observed to have a stabilizing influence in ?.

L791: Are consistent. 'consistent' is rhetorically distinct from 'not inconsistent' here - the former implies support for a hypothesis, whereas the latter only fails to rule it out. Our meaning is intended to be the second one.

L795: Changes alone cannot. Changed.

L797: As in Brinkerhoff. Fixed.

7.3.1-7.3.4: I appreciate the thorough review of prior work here, but it isn't all really necessary for the discussion that comes after in 7.3.5. I would eliminate these sections and fold in the reference to prior work into 7.3.5 only where they are referenced in comparison to your work. This is fair, and also consistent with the other reviewer's suggestion. We have integrated the references into 7.3.5.

L838: Modelling. Fixed.

L893: Is this on one processor? Would make more sense to use an actual computational unit, not just a "laptop" (also specifying the hardware). While individual simulations are run on a single core in this work, the majority of computation is spent doing trivially-parallelized things like randomized Hessian-Vector products and Monte Carlo simulations. We have added the following: For reference, the complete computational pipeline for this work took around 20 hours using one 8-core i9-13900HX processor with 8 performance cores. The bulk of this time ( 60%) was spent computing Hessian-Vector products, while  20% was spend computing the projection ensemble. Both of these tasks are embarrassingly parallel and can be accelerated with additional computing resources.

L911: "This could constitute the largest removal of park lands in the history of the National Park System." This is a striking statement that should go in the abstract. While we agree that it is extremely interesting,

we also think it's not necessarily a primary scientific result, and to us feels out of place when placed so prominently as in the abstract.

L920: The problem. We type the word 'probability' too much, hence the 'probablem'.